



# Detecting segmental anisotropic diffusion in disordered proteins by cross-correlated spin-relaxation

Clemens Kauffmann[1,*], Irene Ceccolini[1,*], Georg Kontaxis[1], and Robert Konrat[1]

[1]Department of Structural and Computational Biology, Max Perutz Laboratories, University of Vienna, Vienna Biocenter Campus 5, A-1030 Vienna, Austria
[*]These authors contributed equally to this work.

**Correspondence:** Robert Konrat (robert.konrat@univie.ac.at)

**Abstract.** Among the numerous contributions of Geoffrey Bodenhausen to NMR spectroscopy his developments in the field of spin-relaxation methodology and theory will definitely have a long lasting impact. Starting with his seminal contributions to the excitation of multiple-quantum coherences he and his group thoroughly investigated the intricate relaxation properties of these "forbidden fruits" and developed experimental techniques to reveal the relevance of previously largely ignored cross-correlated relaxation (CCR) effects, as "the essential is invisible to the eyes". Here we want to discuss CCR within the challenging context of intrinsically disordered proteins (IDPs) and emphasize its potential and relevance for the studies of structural dynamics of IDPs in the future years to come. Conventionally, dynamics of globularly folded proteins are modeled and understood as deviations from otherwise rigid structures tumbling in solution. However, with increasing protein flexibility, as observed for IDPs, this apparent dichotomy between structure and dynamics becomes blurred. Although complex dynamics and ensemble averaging might impair the extraction of mechanistic details even further, spin-relaxation uniquely encodes a protein's structural memory, i.e. the temporal persistence of concerted motions and structural arrangements. Due to significant methodological developments, such as high-dimensional non-uniform sampling techniques, spin-relaxation in IDPs can now be monitored in unprecedented resolution. Not embedded within a rigid globular fold, conventional $^{15}$N spin probes might not suffice to capture the inherently local nature of IDP dynamics. To better describe and understand possible segmental motions of IDPs, we propose an experimental approach to detect the signature of diffusion anisotropy by quantifying cross-correlated spin relaxation of individual $^{15}$N$^1$H$^N$ and $^{13}$C'$^{13}$C$^\alpha$ spin pairs. By adapting Geoffrey Bodenhausen's symmetrical reconversion principle to obtain zero frequency spectral density values we can define and demonstrate more sensitive means to characterize segmental anisotropic diffusion in IDPs.

## 1 Introduction

Geoffrey Bodenhausen's 70th anniversary marks an ideal occasion to take a fresh look at some of his numerous contributions to spin-relaxation methodology and theory. By considering his experiments within the challenging context of intrinsically disordered proteins (IDPs), we want to emphasize their potential and relevance in the future years to come. Arguably, this rediscovery might require some collective effort, as current trends appear to point in the opposite direction. As Paul Schanda put it recently: 'the popularity of detailed spin-relaxation measurements in liquids, *en vogue* 10 or 20 years ago, is declining;





[...] Even with lengthy measurements it is not easy to gain much more insight than "loops are more flexible than secondary structures", which often does not answer mechanistic questions.'(Schanda, 2019, p. 3-4). While intentionally exaggerated, this statement does point to some of the inherent limitations of relaxation experiments. Owing to their convoluted nature, spin-relaxation reports on protein dynamics only in ambiguous terms. A variety of stochastic processes can lead to time correlation functions (TCFs) of identical shape and form(Richert and Blumen, 1994, p. 1-7). In addition, the TCF is not probed directly,

only its spectral density i.e. its Fourier transform is sampled at few select frequencies. Thus, with far more detailed structural models at hand, protein dynamics might appear to be little more than perturbations of otherwise rigid bodies tumbling in solution(Lipari and Szabo, 1982; Halle and Wennerström, 1981; Clore et al., 1990; Halle, 2009). Relaxation experiments commonly employed to calculate protein structures, such as Nuclear Overhauser effects (NOEs) and paramagnetic relaxation enhancements (PREs), are usually modeled without accounting for their dynamic nature(Iwahara et al., 2004; Clore and Iwa-

hara, 2009; Xue et al., 2009; Vögeli, 2014). In a sense, protein dynamics appear separate from protein structure, at least within the structure-function-paradigm.

However, with increasing protein flexibility this apparent dichotomy becomes blurred as structure and dynamics can no longer be considered independent of each other. While complex dynamics and ensemble averaging obfuscate mechanistic details even further, the structural information content of relaxation parameters becomes increasingly apparent. In compari-

son to simple population averaged quantities, such as chemical shifts or scalar couplings, spin-relaxation uniquely encodes a system's structural memory, i.e. the temporal persistence of concerted motions and structural arrangements. Somewhat counter-intuitively, spin-relaxation experiments are among the prime sources of structural information available for disordered systems. However, due to a general lack of analytical descriptions for IDP dynamics(Modig and Poulsen, 2008; Idiyatullin et al., 2001; Bussell and Eliezer, 2001; Kaděřávek et al., 2014; Khan et al., 2015), this notion has been of somewhat academic nature un-

til the recent past. Continuous developments in molecular dynamics (MD) simulation protocols(Piana et al., 2015; Rauscher et al., 2015; Robustelli et al., 2018; Zerze et al., 2019; Piana et al., 2020; Gopal et al., 2021; Shea et al., 2021) demonstrate how this gap can finally be bridged, allowing us to validate, refine and/or analyze dynamic ensemble representations of proteins(Kämpf et al., 2018; Kümmerer et al., 2020; Salvi et al., 2016, 2017). With the necessary timescales becoming increasingly accessible(Stone et al., 2007, 2010; Salomon-Ferrer et al., 2013; Eastman et al., 2017) and the spectral resolution provided by

high-dimensional NUS experiments to overcome the problem of severe spectral overlap(Grudziąż et al., 2018), spin-relaxation in IDPs can be investigated in unprecedented fashion.

This aspect alone suggests a systematic reassessment and evaluation of less commonly employed experiments. Far more pressing, in our opinion, is the inherently local nature of spin-relaxation in IDPs. In contrast to folded proteins, spins in IDPs are not embedded within a fixed molecular tumbling frame. Thus, a single $^{15}$N nucleus per residue as a dynamic probe

probe might not suffice to capture the underlying motions in adequate detail. While detecting and quantifying the presence of anisotropy in IDP dynamics might seem like a rather academic endeavor, it represents an important stepping stone towards the structural interpretation of other experiments. As we recently demonstrated, an appropriate estimate for the average correlation time is an important prerequisite for the angular evaluation of cross-correlated relaxation (CCR) of remote spins(Kauffmann et al., 2021).





More immediate in its structural implications would be the presence of diffusion anisotropy, which has been hypothesized to be of substantial size even in highly disordered proteins such as $\alpha$-Synuclein. Specifically, segmental tumbling of $\alpha$-helical and extended chain conformations has been implied to lead to pronounced diffusion anisotropy effects for intraresidual and sequential $^1$H-$^1$H NOEs(Ying et al., 2014; Mantsyzov et al., 2014, 2015). At the same time, the 3D GAF model(Bremi and Brüschweiler, 1997; Lienin et al., 1998) has been invoked to further rationalize the presence of anisotropy on the local scale of

the peptide plane. This model has recently been reframed by Salvi et al. to analyze MD-simulated $^{15}$N relaxation of a partially disordered protein(Salvi et al., 2017). In essence, it was demonstrated that NH$^N$-TCFs are well-described by the C$^\alpha$C$^\alpha$-TCFs of the same peptide plane as long as variations of the flanking dihedral angles and NH$^N$-librations are accounted for. Explicit corrections for possible diffusion anisotropy effects were not required. However, noticeable deviations could be observed for the transverse $^{15}$N relaxation of the slowly moving residual $\alpha$-helix. Marcellini et al. have reported pronounced diffusion

anistropy within the $\alpha$-helical region of an otherwise disordered construct. Flexible residues were affected noticeably less. It was suggested this might be due to their average orientation in the molecular tumbling frame(Marcellini et al., 2020). The SRLS model of Meirovitch, Freed et al.(Tugarinov et al., 2001; Meirovitch et al., 2006) also predicts pronounced anisotropy for $\alpha$-helices and $\beta$-sheets. However, loops and terminal chain segments appear isotropic, asserting that proteins with substantial internal mobility are best represented by an isotropic global diffusion tensor(Zerbetto et al., 2011).

Arguably, this somewhat ambiguous body of evidence illustrates the inevitable difficulties that come with extending models of folded proteins to IDPs. In fact, many of the above observations might very well be case-dependent. In the present study, we want to approach this question in a more agnostic manner. Are there experimental way to better detect the signature of diffusion anisotropy in IDPs? At what level of evidence could we evoke the mental image of extended chains and $\alpha$-helical segments tumbling in solution? The principal difficulty in characterizing these structural elements lies in their translational periodicity.

In an $\alpha$-helix, NH$^N$ vectors are strongly aligned along the main axis, while in an extended chain, they are oriented perpendicularly. In order to detect orientational biases in the relaxation behavior, additional spin probes with different orientations must be considered. While C$^\alpha$H$^\alpha$ might be suitable for $\alpha$-helices(Barnes et al., 2019), its orientation is too similar to NH$^N$ in the extended chain conformation. Moreover, since it does not share a peptide plane with NH$^N$ it varies as a function of $\phi$ or $\psi$, same as the $^1$H-$^1$H intraresidual and sequential NOEs. Spin probes with less ambiguous orientations would certainly be prefer-

able. For IDPs in particular, Kadeřávek et al. have shown that the NH$^N$ spectral density is best mapped by a combination of transversal and longitudinal CCR rates(Kadeřávek et al., 2014) employing Geoffrey Bodenhausen's symmetrical reconversion principle(Pelupessy et al., 2003, 2007). Together with Bodenhausen and coworkers, this concept was later extended to measure the zero frequency spectral density in a single experiment(Kadeřávek et al., 2015). By translating these concepts to the C'C$^\alpha$ spin pair, we want to derive and demonstrate more sensitive means to detect segmental anisotropic diffusion in IDPs.

## 2    Theory

Our aim is to define an experimental measure for anisotropic diffusion in IDPs. Specifically, we assume anisotropic tumbling of extended chain and $\alpha$-helical segments sufficiently persistent to result in observable spin-relaxation. Before considering





specific experimental aspects, we start by defining the spectral density. Sampled at zero frequency and/or (combinations of) the involved Larmor frequencies, it constitutes the fundamental quantity of all spin-relaxation experiments:

$$J_{\boldsymbol{u},\boldsymbol{v}}(\omega) = \int_0^\infty C_{\boldsymbol{u},\boldsymbol{v}}(t)\cos(\omega t)dt \tag{1}$$

with the time correlation function (TCF),

$$C_{\boldsymbol{u},\boldsymbol{v}}(t) = \langle P_2(\boldsymbol{u}(0)\cdot\boldsymbol{v}(t))\rangle \tag{2}$$

where $P_2(x) = 1.5x^2 - 0.5$ is the second order Legendre polynomial, $\boldsymbol{u}$ and $\boldsymbol{v}$ represent either dipolar unit vectors or principal components of chemical shift anisotropy (CSA) tensors. Note that our simplified definition of the TCF implicitly assumes that time-dependent distance fluctuations factorize and can thus be absorbed into constant coefficients. This requirement will be well-satisfied for the spins considered henceforth.

For most processes, the TCF can be described as a sum/distribution of exponential decays(Lipari and Szabo, 1982; Idiyatullin et al., 2001; Modig and Poulsen, 2008; Khan et al., 2015):

$$C_{\boldsymbol{u},\boldsymbol{v}}(t) = \sum_{k=0}^N a_k e^{-t/\tau_k} \tag{3}$$

Evaluating at $t = 0$ yields a type of normalization condition,

$$\sum_{k=0}^N a_k = C_{\boldsymbol{u},\boldsymbol{v}}(0) = \langle P_2(\boldsymbol{u}(0)\cdot\boldsymbol{v}(0))\rangle \tag{4}$$

which equates to 1 for the familiar case of auto-correlation ($\boldsymbol{u} = \boldsymbol{v}$). For cross-correlation ($\boldsymbol{u} \neq \boldsymbol{v}$), Eq. (4) is bounded within $[-0.5, 1]$.

The spectral density of Eq. (3) is a sum of Lorentzians

$$J_{\boldsymbol{u},\boldsymbol{v}}(\omega) = \sum_{k=0}^N a_k \frac{\tau_k}{1 + (\omega\tau_k)^2} \tag{5}$$

Note that, depending on how the TCF and the spectral density are defined, Eq. (5) might come with additional coefficients such as the familiar factor of $\frac{2}{5}$(Lipari and Szabo, 1982). We prefer the above definitions as they highlight $J_{\boldsymbol{u},\boldsymbol{v}}(\omega)$ as a weighted average. At zero frequency all $\tau_k$ are weighted equally, i.e. $J(0)$ encodes the average correlation time. With increased frequency the impact of larger $\tau_k$ becomes less pronounced. This is illustrated in Fig. 1 for a selection of Larmor frequencies assuming a magnetic field strength of 18.8 T (800 MHz proton Larmor frequency).

Detecting anisotropy amounts to quantifying orientational biases reflected in the $a_k$ and $\tau_k$. Here, we attribute these biases to the relative orientation in extended chain and $\alpha$-helical segments. These structural elements are well-described by an axially symmetric diffusion tensor, which yields the following expression for the spectral density(Tjandra et al., 1996; Woessner, 1962):

$$J_{\boldsymbol{u},\boldsymbol{v}}(\omega) = \sum_{k=0}^2 A_k(\boldsymbol{u},\boldsymbol{v}) \frac{\tau_k}{1 + (\omega\tau_k)^2} \tag{6}$$





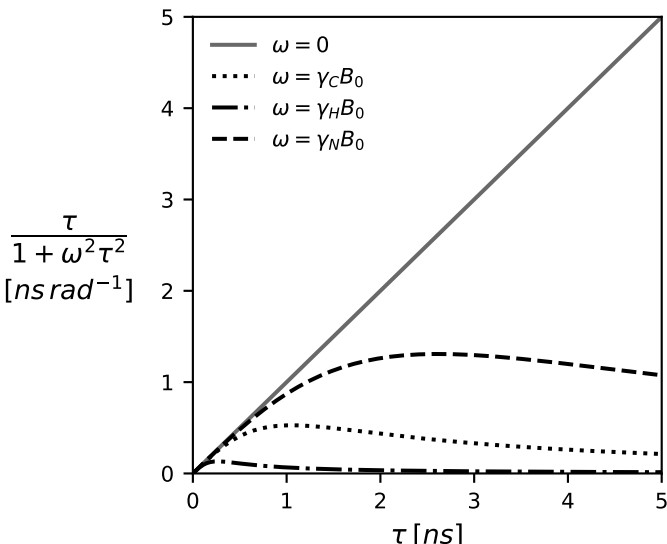

**Figure 1.** The Lorentzian as a function of the correlation time $\tau$ and the (Larmor) frequency. The spectral density $J(\omega)$, which is modeled as a linear combination of Lorentzians, can be pictured as a weighted average of all correlation times $\tau$. Since $J(0)$ weights all correlation times equally, it represents the component most sensitive to correlation times $> 1$ ns. The magnetic field $B_0$ is 18.8 T (800 MHz proton Larmor frequency).

where

$$a_0 \equiv A_0(\boldsymbol{u}, \boldsymbol{v}) = P_2(\theta_{\boldsymbol{u}}) P_2(\theta_{\boldsymbol{v}})$$
$$a_1 \equiv A_1(\boldsymbol{u}, \boldsymbol{v}) = 0.75 \sin(2\theta_{\boldsymbol{u}}) \sin(2\theta_{\boldsymbol{v}}) \cos(\phi_{\boldsymbol{u}} - \phi_{\boldsymbol{v}}) \tag{7}$$
$$a_2 \equiv A_2(\boldsymbol{u}, \boldsymbol{v}) = 0.75 \sin^2(\theta_{\boldsymbol{u}}) \sin^2(\theta_{\boldsymbol{v}}) \cos(2\phi_{\boldsymbol{u}} - 2\phi_{\boldsymbol{v}})$$

and $(\theta, \phi)$ denote the polar angles in the tumbling frame. The $\tau_k$ correspond to the inverted eigenvalues of the axially symmetric diffusion tensor:

$$\tau_k = (6D_\perp + k^2(D_\parallel - D_\perp))^{-1} = D_\perp^{-1}(6 + k^2(\frac{D_\parallel}{D_\perp} - 1))^{-1} \tag{8}$$

with $k = 0, 1, 2$. At this stage, Eq. (6) would not allow us to distinguish between size effects in $\tau_k$ (i.e. segment length) and orientational biases in $A_k(\boldsymbol{u}, \boldsymbol{v})$ (i.e. secondary structure). To quantify $\frac{D_\parallel}{D_\perp}$ only, we consider another interaction described by

a second set of vectors $(\boldsymbol{x}, \boldsymbol{y})$ embedded with different orientations in the same tumbling frame and focus our attention on $J(0)$ for two specific reasons. First and foremost, $J(0)$ is the component most sensitive to the $\tau_k \geq 1$ ns (cf. Fig. 1) commonly associated with tumbling motions(Kämpf et al., 2018). Secondly, $J(0)$ allows us to define a convenient ratio,

$$\frac{J_{\boldsymbol{u},\boldsymbol{v}}(0)}{J_{\boldsymbol{x},\boldsymbol{y}}(0)} = \frac{D_\perp^{-1} \sum_{k=0}^{2} A_k(\boldsymbol{u}, \boldsymbol{v})(6 + k^2(\frac{D_\parallel}{D_\perp} - 1))^{-1}}{D_\perp^{-1} \sum_{k=0}^{2} A_k(\boldsymbol{x}, \boldsymbol{y})(6 + k^2(\frac{D_\parallel}{D_\perp} - 1))^{-1}} = \frac{\sum_{k=0}^{2} A_k(\boldsymbol{u}, \boldsymbol{v})(6 + k^2(\frac{D_\parallel}{D_\perp} - 1))^{-1}}{\sum_{k=0}^{2} A_k(\boldsymbol{x}, \boldsymbol{y})(6 + k^2(\frac{D_\parallel}{D_\perp} - 1))^{-1}} \tag{9}$$



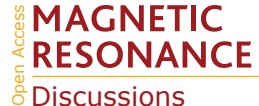

such that the explicit size dependency cancels out. Of course, Eq. (9) still builds on the simplified model of a rigid rotor.

However, presuming that chain- or helix-like structural arrangements are sufficiently stable, we might detect the remnants of Eq. (9) even in case of pronounced local dynamics.

Generally speaking, the effects of fast internal motions are not straightforward to model. The commonly employed model-free (MF)(Lipari and Szabo, 1982) and extended MF(Clore et al., 1990) approaches are not strictly applicable in case of anisotropic diffusion(Daragan and Mayo, 1999; Halle, 2009). For cross-correlated relaxation (CCR) in particular, the proposed

models become quite intricate(Vögeli, 2010). Geoffrey Bodenhausen and coworkers have investigated this topic in a series of publications(Deschamps and Bodenhausen, 2001; Deschamps, 2002; Vugmeyster et al., 2004; Abergel and Bodenhausen, 2004, 2005; Nodet et al., 2008). Here, we model the presence of fast and isotropic motions as simply as possible by introducing a fourth Lorentzian:

$$J_{\boldsymbol{u},\boldsymbol{v}}(\omega) = S^2 \sum_{k=0}^{2} A_k(\boldsymbol{u},\boldsymbol{v}) \frac{\tau_k}{1+(\omega\tau_k)^2} + (1-S^2)P_2(\boldsymbol{u}\cdot\boldsymbol{v})\frac{\tau_3}{1+(\omega\tau_3)^2} \qquad (10)$$

with $\tau_3^{-1} = \tau_{int}^{-1} + 4D_\perp + 2D_\parallel$, where $\tau_{int}$ is the average correlation time of the fast internal motion. The order parameter $S^2 \in [0,1]$ acts as a weight balancing the contributions of slow anisotropic tumbling and fast isotropic motions. To account for the angular relation between $\boldsymbol{u}$ and $\boldsymbol{v}$, $a_3$ necessarily corresponds to $P_2(\boldsymbol{u}\cdot\boldsymbol{v})$, which follows intuitively from condition (4) assuming a fixed angle between $\boldsymbol{u}$ and $\boldsymbol{v}$.

Of course, the additional Lorentzian can be rationalized in terms of existing models. Following Bax and coworkers(Barbato

et al., 1992; Tjandra et al., 1995), the factorization of global tumbling and internal motions assumed in the MF approach(Lipari and Szabo, 1982) is modeled by coupling the internal motions to an effective overall tumbling time $\tau_{eff} = (4D_\perp + 2D_\parallel)^{-1}$ which yields $\tau_3^{-1} = \tau_{int}^{-1} + 4D_\perp + 2D_\parallel$. This approach contrasts the introduction of explicit cross-terms(Kroenke et al., 1998) which retain their orientational biases even in the limit $S^2 = 0$. In the MF approach of Halle(Halle and Wennerström, 1981), which allows for the presence of diffusion anisotropy, local motions are modeled by simply adding a $(1-S^2)$-weighted TCF

of further unspecified form(Halle, 2009). In this picture, we represent the internal motions by a mono-exponential decay with $\tau_3$ following from the considerations above. The familiar interval of $S^2 \in [0,1]$, which applies for auto-correlated TCFs only, corresponds to the MF adaptation of Kroenke et al.(Kroenke et al., 1998). With an effective isotropic tumbling time $\tau_{eff}$ coupling to the internal motions, the equivalence $a_3 = \sum_{k=0}^{2} A_k(\boldsymbol{u}\cdot\boldsymbol{v}) = P_2(\boldsymbol{u}\cdot\boldsymbol{v})$ is obtained. The same expressions can be derived from the common approximation for the CCR order parameter $S_{uv}^2 = S^2 P_2(\boldsymbol{u}\cdot\boldsymbol{v})$(Daragan and Mayo, 1996; Ghose

et al., 1998). Then, the decay of the internal TCF towards its asymptotic value $S_{uv}^2$ is encoded by the factor $P_2(\boldsymbol{u}\cdot\boldsymbol{v}) - S_{uv}^2 = (1-S^2)P_2(\boldsymbol{u}\cdot\boldsymbol{v})$. In principle, this approximation can be extended to model the entire cross-correlated TCF(Tjandra et al., 1996; Halle, 2009). With the angular dependencies available in explicit form, we see no reason to simplify any further.

While the fast isotropic motions could be modeled in more detail to better fit the shape of the TCF using e.g. the extended MF approach(Clore et al., 1990) or correlation time distributions(Hsu et al., 2018), we only intend to divide $J(0)$, i.e. the

TCF's enclosed area, into contributions with and without orientational biases. More impactful is the assumption of equally weighted isotropic motions for $(\boldsymbol{u},\boldsymbol{v})$ and $(\boldsymbol{x},\boldsymbol{y})$. This simplification is introduced primarily to keep the amount of parameters manageable. It assumes that order parameters of the same peptide plane should be reasonably comparable (Chang and Tjandra,





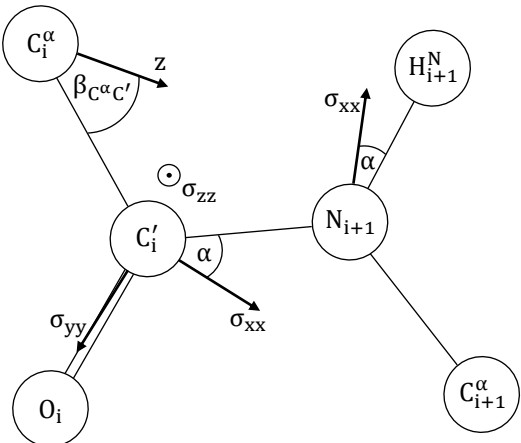

**Figure 2.** The peptide plane as defined by Corey and Pauling(Corey et al., 1953): $C^\alpha$–C' = 1.53 Å, C'–O = 1.24 Å, C'–N = 1.32 Å, N–$C^\alpha$ = 1.47 Å. $C^\alpha$-C'-O = 121°, $C^\alpha$-C'-N = 114°, O-C'-N = 125°, C'-N-H = 123°, C'-N-$C^\alpha$ = 123°, H-N-$C^\alpha$ = 114°. N–H = 1.04 Å is taken from Ottiger and Bax (Ottiger and Bax, 1998). The $^{15}$N and $^{13}$C' CSA principal components are adapted from Bodenhausen and coworkers(Cisnetti et al., 2004; Loth et al., 2005). $^{15}$N: $\Delta_N \approx \sigma_{xx} - \sigma_{yy} \approx \sigma_{xx} - \sigma_{zz}$ = 170 ppm, $\alpha$ = 20°. $^{13}$C': $\sigma_{xx}$ = 249.4 ppm, $\sigma_{zz}$ = 87.9 ppm, $\alpha$ = 37°. $\sigma_{yy}$ = 191.1 ppm is obtained from the average chemical shift of Ubiquitin (BMRB 17769)(Cornilescu et al., 1998)) following the suggested calibration(Cisnetti et al., 2004). The main axis $z$ of the diffusion anisotropy tensor is assumed to lie in the peptide plane. Its orientation is encoded by $\beta_{C^\alpha C'}$

.

2005; Ferrage et al., 2006; Wang et al., 2006; Salvi et al., 2017). Differences in local mobility between $(\boldsymbol{u}, \boldsymbol{v})$ and $(\boldsymbol{x}, \boldsymbol{y})$ will necessarily result in systematic deviations from the implied isotropic baseline, with $D_\parallel = D_\perp = D$,

$$\frac{J_{\boldsymbol{u},\boldsymbol{v}}(0)}{J_{\boldsymbol{x},\boldsymbol{y}}(0)} = \frac{P_2(\boldsymbol{u} \cdot \boldsymbol{v})(S^2(6D)^{-1} + (1-S^2)\tau_3)}{P_2(\boldsymbol{x} \cdot \boldsymbol{y})(S^2(6D)^{-1} + (1-S^2)\tau_3)} = \frac{P_2(\boldsymbol{u} \cdot \boldsymbol{v})}{P_2(\boldsymbol{x} \cdot \boldsymbol{y})} \tag{11}$$

From Eq. (11), it can be seen that the ratio (9) encodes a simple and intuitive balance: Isotropic motions tend towards the limit (11), while anisotropic motions deviate from it. To explore the extent of this effect in the presumed case of segmental tumbling of $\alpha$-helices and extended chains, we need to consider the abstract notion of $J_{\boldsymbol{u},\boldsymbol{v}}(0)$ and $J_{\boldsymbol{x},\boldsymbol{y}}(0)$ from an experimental perspective.

## 3 Methods

We will assume the canonical peptide plane geometry of Corey and Pauling(Corey et al., 1953) as depicted in Fig. 2 including approximate principal components of the CSA tensors for $^{15}$N and $^{13}$C' adapted from Geoffrey Bodenhausen and coworkers(Cisnetti et al., 2004; Loth et al., 2005).



As demonstrated by Kaděrávek et al.(Kaděrávek et al., 2014), the spectral densities of IDPs are best mapped by combining

the transversal ($\Gamma_{xy}$) and the longitudinal ($\Gamma_z$) CCR rates between the $^{15}$N CSA and the NH$^N$ dipole. Employing the notation of Bodenhausen and coworkers(Cisnetti et al., 2004), we have

$$\Gamma_{xy}^{N/NH} = k_{N/NH}\Delta_N\left[4J_{NH,xx}(0) + 3J_{NH,xx}(\omega_N)\right] \tag{12}$$

$$\Gamma_z^{N/NH} = k_{N/NH}\Delta_N\left[6J_{NH,xx}(\omega_N)\right] \tag{13}$$

$$k_{N/NH} = \frac{2}{5}\frac{1}{24\pi}\frac{\mu_0\hbar\gamma_n\gamma_h}{r_{NH}^3}B_0\gamma_n$$

where $\mu_0$ is the vacuum permeability, $\hbar$ is the reduced Planck constant, $\gamma$ is the gyromagnetic ratio, $r$ is the distance between the nuclei, $B_0$ is the magnetic field strength and $\Delta_N = (\sigma_{xx} - \sigma_{yy}) = (\sigma_{xx} - \sigma_{zz})$ is the size difference of the $^{15}$N CSA principal components (in ppm). Mapping $J_{NH,xx}(0)$ amounts to the simple subtraction $\Gamma_{xy}^{N/NH} - 0.5\Gamma_z^{N/NH}$.

To complement these rates, we consider their counterparts for the $^{13}$C' CSA and the C'C$^\alpha$ dipole.:

$$\Gamma_{xy}^{C'/C'C^\alpha} = k_{C'/C'C^\alpha}\left[(\sigma_{xx} - \sigma_{zz})(4J_{C'C^\alpha,xx}(0) + 3J_{C'C^\alpha,xx}(\omega_C)) + (\sigma_{yy} - \sigma_{zz})(4J_{C'C^\alpha,yy}(0) + 3J_{C'C^\alpha,yy}(\omega_C))\right] \tag{14}$$

$$\Gamma_z^{C'/C'C^\alpha} = k_{C'/C'C^\alpha}\left[(\sigma_{yy} - \sigma_{zz})6J_{C'C^\alpha xx}(\omega_C) + (\sigma_{yy} - \sigma_{zz})6J_{C'C^\alpha yy}(\omega_C)\right] \tag{15}$$

$$k_{C'/C'C^\alpha} = \frac{2}{5}\frac{1}{24\pi}\frac{\mu_0\hbar\gamma_c^2}{r_{C'C^\alpha}^3}B_0\gamma_c$$

with $xx,yy,zz$ referring to the principal components of the fully anisotropic $^{13}$C' CSA tensor. Again, high frequency contributions can be eliminated via the linear combination $\Gamma_{xy}^{C'/C'C^\alpha} - 0.5\Gamma_z^{C'/C'C^\alpha}$.

While the measurement of transverse CCR is well-established, longitudinal CCR has been studied considerably less. In

part this is due to subtleties of the involved relaxation pathways which involve multi-exponential cross- and cross-correlated relaxation effects. Another reason lies in technical limitations as longitudinal relaxation rates are generally smaller due to their Larmor frequency dependence. Notably, this effect is far less pronounced for the smaller correlation times present in IDPs (cf. Figure 1). While $^{15}$N$^1$H$^N$ relaxation is well understood and several sensitive NMR techniques have been proposed to measure $\Gamma_{xy}^{N/NH}$ and $\Gamma_z^{N/NH}$(Tjandra et al., 1996; Kroenke et al., 1998; Pelupessy et al., 2003, 2007; Kaděrávek et al., 2015), $^{13}$C'

relaxation is generally more problematic(Dayie and Wagner, 1997; Wang et al., 2006). Since we could not find any previous attempts to measure the longitudinal CCR rate $\Gamma_z^{C'/C'C^\alpha}$ in the literature, we see fit to assess its general feasibility.

Aside from the symmetrical reconversion principle of Bodenhausen and coworkers(Pelupessy et al., 2003, 2007), $^{13}$C'/$^{13}$C'$^{13}$C$^\alpha$ CSA-DD CCR can be measured either by monitoring the relaxation asymmetry of the $^{13}$C'$^{13}$C$^\alpha$ doublet or by means of a 'quantitative gamma' experiment in which sum and difference of the $^{13}$C' doublet relaxation are measured independently. In

contrast to previous approaches relying on two separate experiments ('reference' and 'cross')(Schwalbe et al., 2002), we determine $\Gamma_z^{C'/C'C^\alpha}$ by quantifying the different longitudinal relaxation in the $^{13}$C'$^{13}$C$^\alpha$ doublet recorded in a non-constant-time C' evolution following the relaxation period. Transverse relaxation $\Gamma_{xy}^{C'/C'C^\alpha}$ is measured by more conventional quantification of differential line broadening of the $^{13}$C'$^{13}$C$^\alpha$ doublet recorded in constant-time mode.

To obtain sufficient spectral resolution the CCR rates are measured directly from the intensity difference in a $^{13}$C$^\alpha$-coupled

3D HNCO experiment; (i) in case of transversal CCR by quantification of differential line broadening of the $^{13}$C'$^{13}$C$^\alpha$ doublet



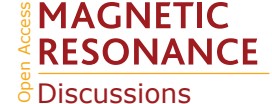

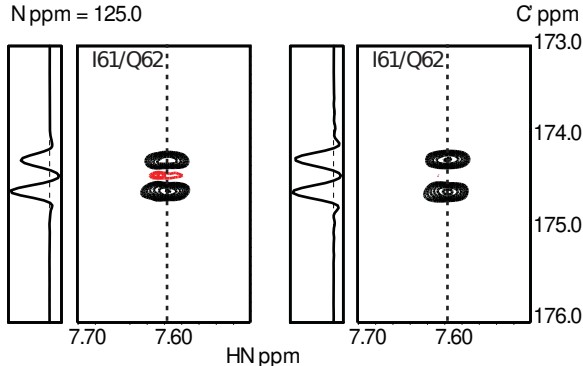

**Figure 3.** Experimental results from the measurements of transverse (A), left) and longitudinal (B), right) CSA-DD cross-correlated relaxation as described in Sect. 3. Data were obtained using Ubiquitin, a small globular protein of 76 residues. The figure shows the spectral region for the peptide plane spanning residues I61/Q62. The asymmetry of the $^{13}C^\alpha$ doublet is visible in the cross-sections taken at the positions indicated by dashed lines. As expected, CCR effects are more pronounced in the case of transverse relaxation.

during constant-time $^{13}C'$ evolution and (ii) for longitudinal CCR during real-time $^{13}C^\alpha$ coupled $^{13}C'$ evolution preceded by a longitudinal relaxation delay $T$ during which $^{13}C'/^{13}C'^{13}C^\alpha$ CSA-DD CCR is active. This approach yields reliable longitudinal CCR rates as long as the mixing time $T$ is short compared to $^{13}C'$ $T_1$ relaxation. Typical data obtained for the small globular protein Ubiquitin are shown in Fig. 3

To suppress $^{13}C'^{13}C^\alpha$ cross-relaxation a $^{13}C'$ is inverted in the middle of the relaxation delay $T$. Additional unwanted CCR pathways involving the $^{13}C'$ CSA and $^{13}C'^1H/^{13}C'^{15}N$ dipoles are suppressed by $^1H$ decoupling and $^{15}N$ inversion. The CCR rates are obtained from the $^{13}C'^{13}C^\alpha$ doublet as $\log(\frac{I_a}{I_b})/2T$. Details of the pulse sequence and NMR parameters will be given elsewhere. Two exemplary $^{13}C'^{13}C^\alpha$ doublets measured for I61/Q62 in human Ubiquitin are shown in Fig. 3.

With the general feasibility of the measurements demonstrated, we can now define a ratio $Q$ analogous to Section 2, Eq. (9):

$$Q \equiv \frac{\Gamma_{xy}^{C'/C'C^\alpha} - 0.5\Gamma_z^{C'/C'C^\alpha}}{\Gamma_{xy}^{N/NH} - 0.5\Gamma_z^{N/NH}} = \frac{4k_{C'/C'C^\alpha}\left[(\sigma_{xx} - \sigma_{zz})J_{C'C^\alpha,xx}(0) + (\sigma_{yy} - \sigma_{zz})J_{C'C^\alpha,yy}(0)\right]}{4k_{N/NH}\Delta_N J_{NH,xx}(0)} \quad (16)$$

To assess the sensitivity of $Q$ (16), it is evaluated according to Eqs. (7), (8),(10), (12-15) with $\tau_3^{-1} = \tau_{int}^{-1} + 4D_\perp + 2D_\parallel = \tau_{int}^{-1} + \tau_{eff}^{-1}$ under the following conditions: As specified in Fig. 2, all CSA tensors have fixed orientation and size. The main axis $z$ of the axially symmetric diffusion tensor is assumed to lie in the peptide plane, hence $Q$ is a function of the polar angles $\theta$ only, see Eq. (7). Defining the $C^\alpha C'$ orientation as 0° reference, the main axis is rotated from 0° to 180° towards the $NH^N$

vector assuming anisotropy values $\frac{D_\parallel}{D_\perp}$ of 1.5 and 2.5, effective tumbling times $\tau_{eff} = (4D_\perp + 2D_\parallel)^{-1}$ of 1 and 2.5 ns, internal correlation times $\tau_{int}$ of 100 and 500 ps and order parameters $S^2$ between 0 an 1. The magnetic field strength $B_0$ is the same for all rates and thus cancels out. The results are summarized in Fig. 4.



# 4 Results and Discussion

Experimental considerations necessarily result in compromises. The fully anisotropic $^{13}$C' CSA not only leads to spectral
density contributions of two perpendicular components, it is also subject to considerable variations(Markwick et al., 2005;
Cisnetti et al., 2004; Loth et al., 2005). One might be tempted to avoid the uncertainties and complications that come with the
$^{13}$C' CSA by considering dipolar relaxation only. However, compared to the NH$^N$ spin pair, the small gyromagnetic ratios and
long internucleic distances of other dipoles lead to far smaller and less sensitive rates(Carlomagno et al., 2000). In addition,
the $J(0)$ components are generally neither dominant nor easily separated. The $^{13}$C' CSA both provides effective means of
relaxation and allows for a straightforward extraction of $J(0)$ components. With an approximate ratio $(\sigma_{xx} - \sigma_{zz})/(\sigma_{yy} - \sigma_{zz}) \approx 1.5$ and the beneficial orientation of the $C'C^\alpha$ vector, the $J_{C'C^\alpha,xx}(\omega)$ contribution is generally far more pronounced:
For $30° \leq \alpha \leq 44°$ the TCF amplitudes would be $0.48 \leq P_2(C'C^\alpha \cdot xx) \leq 0.79$, $0.02 \geq P_2(C'C^\alpha \cdot yy) \geq -0.29$ based on the
geometry of Fig. 2.

Fig. 4 shows the ratio $Q$ (16) for different choices of $\tau_{eff}, \tau_{int}, S^2, \frac{D_\parallel}{D_\perp}$ as a function of the diffusion tensor orientation. The
main axis is assumed to lie in the peptide plane with the orientation denoted relative to $C'C^\alpha$ in terms of the angle $\beta_{C'C^\alpha}$, see
Fig. 2. Comparing all panels (a)-(f) at once, it can be seen that the isotropic ($S^2 = 0$) baseline at around 0.3 is independent of
the correlation times $\tau_{eff}$ and $\tau_{int}$ as derived in Eq. (11). The same value is obtained for $\frac{D_\parallel}{D_\perp} = 1$, which is easily assessed from
the convergence behavior for different anisotropy values in (c),(f) and (a),(b),(d),(e). How strongly $Q$ reports on the asserted
presence of diffusion anisotropy depends on the $S^2$-mediated weight difference between the orientation dependent $A_k\tau_k$ (7)
and the isotropic $\tau_{eff}$. Both higher overall tumbling $\tau_{eff}$ and smaller isotropic motions $\tau_{int}$ yield a more sensitive $Q$ for
increasingly smaller order parameters $S^2$, see panels (a),(b),(d),(e). Note that the particular choice of $\tau_{int}$ and $S^2$ is to some
extent arbitrary as $A_k\tau_k$, $\tau_{int}$ and $S^2$ merely represent the isotropic and anisotropic contributions to the TCF's enclosed area
$J(0)$. Still, $\tau_{eff} \geq 1$ ns $> \tau_{int}$ was chosen based on timescales recently reported for IDPs(Kämpf et al., 2018).

Besides the obvious influence of $\frac{D_\parallel}{D_\perp}$ itself, $Q$ strongly depends on the orientation of the diffusion tensor. Highlighted in all
panels (a)-(f) are the orientations of the NH$^N$ and the C$^\alpha$C$^\alpha$ vectors. In an extended chain, C$^\alpha$C$^\alpha$ is approximately parallel
to the main axis while NH$^N$ stands perpendicular to it or vice versa for an $\alpha$-helix. Both orientations correspond well to the
minimum and maximum of $Q$. The range of $Q$ depends on the size of the anisotropy $\frac{D_\parallel}{D_\perp}$. For a value of 2.5, as was previously
asserted for $\alpha$-Synuclein(Mantsyzov et al., 2014, 2015), the effect on $Q$ can be quite substantial, panels (a),(b),(d),(e). For $\frac{D_\parallel}{D_\perp}$
= 1.5, it is far less pronounced, panels (c),(f).

We conclude that, if the concept of anisotropic diffusion of segmental $\alpha$-helices and extended chains is reasonably applicable
and sufficiently pronounced, $Q$ would allow to unambiguously detect its signature. Actual quantification of $\frac{D_\parallel}{D_\perp}$ would of
course be obstructed by many unknown variables and experimental uncertainties as well as the limited validity of the asserted
dynamic model. While the presence of relaxation-active tumbling motions do imply a certain degree of local rigidity, the
structural heterogeneity of IDPs certainly challenges many of the commonly made assumptions. Still, the ratio $Q$ might give
an indication of how reasonable these concepts are for a given protein system. While particularly sensitive to large correlation
times, $Q$ will report on all sources of anisotropy present in $J(0)$. Differences in local mobility, CSA tensor variations, overall





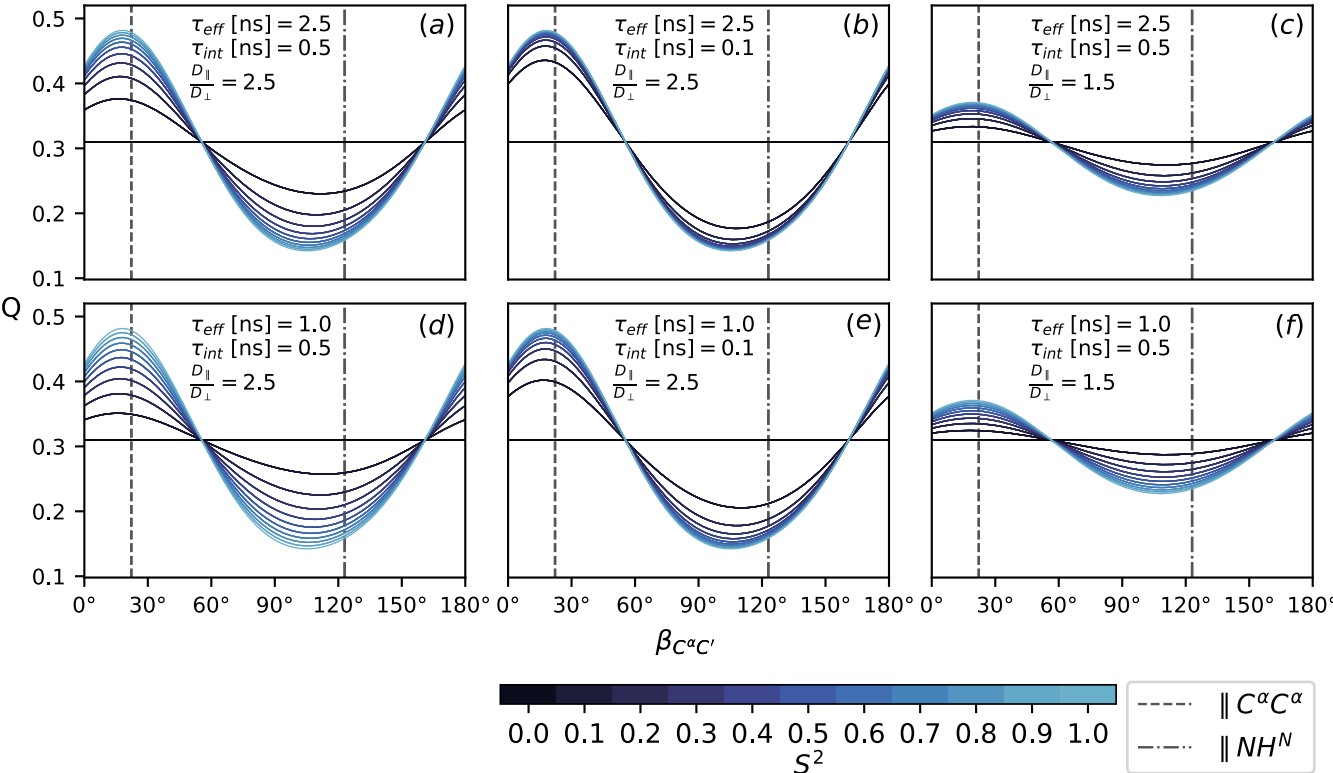

**Figure 4.** The ratio $Q$, Eq. (16), as a function of the diffusion tensor orientation denoted by $\beta_{C^\alpha C'}$, Fig. 2. Dashed lines indicate the orientation of the $NH^N$ and the $C^\alpha C^\alpha$ vector. All rates are calculated according to Eqs. (7), (8),(10), (12-15) with $\tau_3^{-1} = \tau_{int}^{-1} + \tau_{eff}^{-1}$. Order parameters from 0 and 1 are color-coded. Panels (a)-(f) show $Q$ for different choices of $\tau_{eff}, \tau_{int}$ and $\frac{D_\parallel}{D_\perp}$. The magnetic field strength $B_0$ is the same for all rates.

structural flexibility and experimental uncertainties will certainly shift and blur the ratio expected for isotropic motions. Still, if we assume a set of consecutive residues to experience shared anisotropic diffusion, $Q$ should exhibit a systematic and sequence-persistent pattern.

In addition, the spectral densities can always be evaluated directly. While the proposed experiments do not allow to map $J_{C'C^\alpha,xx}$ and $J_{C'C^\alpha,yy}$ individually, the contributions of different Larmor frequencies are fully separated. Graphical representations in particular can provide model-independent intuition about the timescales at play(Idiyatullin et al., 2001; Křížová et al., 2004; Kaděřávek et al., 2014). Expressions such as $J(0) - J(\omega)$(Idiyatullin et al., 2001), intended to suppress the contribution of faster timescales (cf. Fig. 1), are available as well. While introduced and justified primarily in terms of diffusion anisotropy,

we expect the combination of transversal and longitudinal $C'/C^\alpha C'$ CCR rates to prove informative even outside the scope considered here. For the locally dominated dynamics of IDPs in particular, differences and similarities to the $NH^N$ spectral density can provide valuable structural insights even without invoking specific dynamic models(Kämpf et al., 2018).



# 5 Conclusions

On the occasion of Geoffrey Bodenhausen's 70th anniversary, we built on his extensive body of work to conceptualize ex-
perimental means for the investigation of segmental anisotropic tumbling in IDPs. Spectral density mapping protocols based
on transversal and longitudinal CCR of $NH^N$ were translated to the $C^\alpha C'$ spin pair of the same peptide plane. By isolating
and comparing the zero frequency contributions we derived an intuitive experimental measure for the presence of anisotropic
dynamics. Provided that model-free assumptions are applicable, we show that pronounced anisotropic tumbling of extended
chain and $\alpha$-helical segments should be readily detectable. But even outside the context of conventional dynamic models, con-
tributions of different frequencies can be separated and assessed similarly to spectral density mapping protocols. Interestingly,
the required measurement of longitudinal $C'/C^\alpha C'$ CCR has not been investigated before. Hence, a simple proof of concept for
a possible measurement scheme was provided. To further substantiate and explore the presented concepts in an experimental
setting, a systematic evaluation of different pulse sequences is currently under preparation in our lab.

While detecting and quantifying the presence of anisotropy in IDP dynamics might seem like a humble academic endeavor,
we believe this to be an important step not only towards a better understanding of this important protein family but also
towards immediate applications in biological and biomedical research as well as drug design. We thus take particular delight
from the fact that Geoffrey's *l'art pour l'art* pulse sequence design is also a telling testimony for the unforeseeable impact of
non-utilitarian basic research driven and inspired by scholarly thinking.

*Author contributions.* RK, CK and IC devised the project. CK compiled the theoretical considerations. IC and CK performed the numerical
simulations. GK and RK assessed the experimental feasibility. GK designed the pulse sequence. IC and CK illustrated the results. CK wrote
the manuscript with contributions from all authors.

*Competing interests.* No competing interests are declared.

*Acknowledgements.* This work was supported by the Austrian Science Fund FWF (P28359 and P28937).





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
