# Peer review of "Detecting anisotropic segmental dynamics in disordered proteins by cross-correlated spin-relaxation"

_Magnetic Resonance, 2021_

## Referee Comment (RC2)

The authors of this manuscript mr-2021-35 suggest a strategy for detecting anisotropic diffusion, $D_\perp \neq D_\parallel$ (where $\boldsymbol{D}$ denotes a second-rank diffusion tensor), in intrinsically disordered proteins (IPDs), on the basis or NMR cross-correlated relaxation (CCR). The spin systems considered include $^{15}$N–$^1$H and $^{13}$C'–$^{13}$C$^\alpha$. It was shown previously for $^{15}$N–$^1$H that the transverse and longitudinal CCR rate constants can be combined linearly so as to yield an expression which depends solely on the measurable spectral density, $J^{CD}(\omega)$, for $\omega = 0$ ("C" stands for the $^{15}$N CSA interaction and "D" for the $^{15}$N–$^1$H dipolar interaction). This strategy is adopted here for the $^{13}$C'–$^{13}$C$^\alpha$ spin system (with "C" standing for the $^{13}$C' CSA interaction and "D" for the $^{13}$C'–$^{13}$C$^\alpha$ dipolar interaction). An improved experimental method for measuring the longitudinal CCR rate constant for $^{13}$C'–$^{13}$C$^\alpha$ is reported. $J^{CD}(0)$ data are acquired experimentally for both spin systems. On the theoretical side a new form for $J^{CD}(\omega)$, which should enable detecting $D_\perp \neq D_\parallel$ when $\omega = 0$, is suggested. Detection sensitivity is illustrated. My comments refer to several matters of general character, and to the form of the $J^{CD}(\omega)$ function.

**General matters**. **(1)** This study is totally predictive in nature. No validation, and/or examples illustrating actual applicability are provided. **(2)** A cursory survey of the NMR-based studies of IDPs cited in mr-2021-35 shows that methods examining local features point to a random-coil situation (e.g., see Mantsyzov et al. **2014**). CCR in $^{15}$N–$^1$H and $^{13}$C'–$^{13}$C$^\alpha$ is a local feature. Obtaining information on long-range "order", in particular "diffusion anisotropy", requires empirical spectral densities comprising statistical elements, to be used in combination with coil libraries and molecular dynamics simulations (e.g., see Mantsyzov et al. **2014**). These elements are absent in mr-2021-35 scheme. **(3)** It is indicated that anisotropic "segmental" motion is

targeted. How are these "segments" defined? In other word, how is the second-rank diffusion tensor, $D$, defined? (**4**) It is pointed out that NMR relaxation analysis methods applicable to folded proteins are not applicable to IDPs. The model-free (MF) spectral density, a variant of which is suggested here, refers to protein and probe as rigid bodies moving in a statistically independent (decoupled) manner. In IDPs the protein is not rigid and its motion is not decoupled from the motion of the probe. The spectral density suggested here does not reflect these features; rather, it is similar in character to the MF spectral density. (**5**) In the context of item (2) – please note that the "dynamics detectors" method (Smith et al. *Angew. Chem. Int. Ed.* **2017** *56*. 13590), shown to actually surpass MF, comprises statistical elements. Recently it was applied to proteins in solution (Smith et al. *JCP* **2019**, *151*, 034102). The authors might want to check applicability to IDPs.

$J^{CD}(\omega)$. Let us focus on $^{15}N-^{1}H$ spin system as paradigm. The essence of the following is equally applicable to the spin system $^{13}C'-^{13}C^{\alpha}$. Within a very good approximation the $^{15}N-^{1}H$ dipolar/$^{15}N$ CSA cross-correlated spectral density, $J^{CD}(\omega)$, is given for globular proteins by (Tjandra et al. *JACS* **1996**, *118*, 6986):

$$J^{CD}(\omega) \cong P_2(\cos\theta)\, J^{DD}(\omega) = P_2(\cos\theta)\, j_0(\omega) \qquad (1)$$

where $\theta$ denotes the angle between the principal axes of the axial $^{15}N-^{1}H$ dipolar and $^{15}N$ CSA tensors. $J^{DD}(\omega)$ is the measurable spectral density for auto-correlated dipolar relaxation. $j_0(\omega)$ is the $K = 0$ component of the solution, $j_K(\omega)$, $K = 0, 1, 2$, of the dynamic model considered. *For wobble-in-a-cone $j_0(\omega)$ is given by the MF spectral density* (Lipari & Szabo *JCP* **1981**, *75*, 2971).

The point I wish to make is as follows. The physical picture underlying NMR relaxation is inherent in the $j_K(\omega)$, $K = 0, 1, 2$, functions, which are not measurable. To render them measurable one has to carry out appropriate frame transformations. The equality in eq 1 is due to the fact that the model-related local ordering frame, M, where the $j_K(\omega)$ functions are defined, and the NMR-related D frame, are taken the same in MF. The approximate equality in eq 1 is due to the fact that the C frame and the M = D frame are not the same but $\theta$ is small (Tjandra et al. *JACS* **1996**, *118*, 6986). Hence, only the $K = 0$ component, $P_2(\cos\theta)$, in the M = D to C frame transformation survives.

Thus, $J^{DD}$, $J^{CC}$ and $J^{CD}$ are geometric implementations of $j_K(\omega)$, $K = 0, 1, 2$. Equation (2) of mr-2021-35 represents time-dependent cross-correlation between the (axial) D and C frames. As shown above, the C frame is obtained from the D (= M) frame by a frame transformation based on time-independent Euler angles. This post-solution frame transformation can also be found in a different but related case in Szabo *JCP* **1980**, *72*, 4620. This invalidates eq 2.

Equations 2–4 do not comply with the theory of moments, which underlies the MF time correlation function utilized here as basis. Equation 6 represents the frame transformation from the $^{15}$N–$^1$H dipolar frame ($u$) to the global diffusion frame (Tjandra et al. *JACS* **1995**, *117*, 12562). One should have:

$$\sum_{K=0,1,2} A_K(u) \frac{\tau_K}{1 + (\omega\tau_K)^2} \tag{2}$$

The index, $v$, in eq 6 [$(A_K(u,v))$] is not comprehensible in this context. Equation 7 is very confusing. The coefficients $A_K(u)$ are in actual fact time-independent trigonometric functions. Yet, the quantities $A_K(u,v)$ in eq 7 of mr-2021-35 feature as coordinates "($\theta$, $\varphi$), which denote the

polar angles in the tumbling frame"; this implies time-dependence of angles alien to the essence of $A_K(u)$.

Based on eqs 2–9, shown above to be problematic, the authors obtain the spectral density:

$$J_{u,v}(\omega) = S^2 \sum_{K=0,1,2} A_K(u,v) \frac{\tau_K}{1+(\omega\tau_K)^2} + (1-S^2) \sum_{K=0,1,2} A_K(u,v) \frac{\tau_3}{1+(\omega\tau_3)^2} \quad (3)$$

where $\tau_3^{-1} = \tau_{int}^{-1} + 4\,D_\perp + 2\,D_\parallel$. The correlation time, $\tau_{int}$, is considered to be the average correlation time for internal motion. The correlation time $\tau_{eff} = (4\,D_\perp + 2\,D_\parallel)^{-1}$ is considered to be the effective correlation time for global tumbling. The two principal values, $D_\perp$ and $D_\parallel$, of the second-rank diffusion tensor correspond to the three eigenvalues of the symmetric top, given by $\tau_K^{-1} = 6\,D_\perp + K^2(D_\parallel - D_\perp)$, $K = 0, 1, 2$. This yields $\tau_0^{-1} = 6\,D_\perp$, $\tau_1^{-1} = D_\parallel + 5\,D_\perp$, and $\tau_2^{-1} = 4D_\parallel + 2\,D_\perp$. Why should the effective correlation time for global motion be equal to $\tau_2^{-1}$? In what sense does the expression $\tau_{int}^{-1} + 4\,D_\perp + 2\,D_\parallel$ represent – as indicated – dynamical coupling, which means concerted time-evolution of the global and local motional degrees of freedom? What physical framework could possibly justify the second term of eq 3?

---

## Referee Comment (RC3)

Reply to the authors' rebuttal to reviewer #2

Regarding general matter **(1)**, as referee #1 has pointed out as well, our study is indeed predictive and conceptual in nature. In our paper, we want to acknowledge Geoffrey Bodenhausen's contributions to the field by discussing how CCR experiments developed in his group over the years can potentially be used to investigate and characterize local (segmental) dynamics in (partially) disordered proteins.

Authors submitting altogether predictive manuscripts to established journals are typically required to demonstrate applicability. It has been assumed that submissions to the Festschrift issue of MR Discussions honoring Geoffrey Bodenhausen are also subject to this requirement.

Let us clarify: We do not claim nor intend to provide a thorough and general description of IDP dynamics. While segmental motions are clearly present in IDPs (Rezaei-Ghaleh et al. 2018, Parigi et al. 2014), it is obvious that the concept of diffusion anisotropy cannot be expected to rigorously apply. Still, this somewhat elusive concept has been invoked in previous studies even for proteins such as $\alpha$-Synuclein (Mantsyzov et al. 2014). This leads us to the central question of our study: If the segmental motion of IDPs exhibited features/trends associated with anisotropic tumbling, how could we best detect it? Can we define a sensitive and unambiguous experimental measure?

The first sentence is obvious and the objective of this work is appreciated. The issues at stake include the definition of the "segment" as conceived by Mantsyzov et al.[1] (to which the authors refer), and the physical relevance of the "measure". $C^\alpha$ and C' belong to residue $i - 1$; N and $H^N$ belong to residue $i$. $D_\parallel/D_\perp$ are likely to differ for consecutive $(i - 1) - i$ pairs. How will then the ratios, $D_\parallel/D_\perp$, between the parallel and perpendicular components of the global diffusion tensor, $\boldsymbol{D}$, of the "segment" be determined? The "measure" suggested is ill-defined – see the original review report, and this report. On the most general level – in the Introduction the authors indicate that time-correlation functions (TCFs) developed for folded proteins, which include model-free-type TCFs, are not applicable to intrinsically disordered proteins (IDPs). Yet, they suggest a TCF of this type. With regard to these matters have I suggested considering – or at least discussing – coil libraries and molecular dynamics simulations, e.g., as in Mantsyzov et al.[1]

Complementing $NH^N$ CCR, we show that $C'C^\alpha$ CCR allows to probe the peptide plane quite literally from a different angle. The isolated zero frequency components are straightforward to compare and particularly sensitive for larger correlation times, allowing to probe the presence or lack(!) of anisotropic dynamics in IDPs on the local scale of the peptide unit. To assess the combined information content of the CCR rates, we build on the previously invoked and simplified model of a tumbling symmetric top. The "dampening" effect of local motions is approximated as simply as possible using a single exponential decay with equal weight for both $NH^N$ and $C'C^\alpha$. We do not claim this treatment is rigorous nor do we imply that features of this MF-like anisotropic tumbling should be expected for all protein systems. The experiments were conceptualized and designed to assess the presence of features/trends associated with this simplified dynamic model.

As presented, the axial top includes a given $(i - 1) - i$ pair. Its global tumbling is represented by the first term of eq 10 of mr-2021-35. That equation is depicted below as eq 1. The "dampening" effect of the local motions is given by the second term or eq 1.

$$J_{u,v}(\omega) = S^2 \sum_{K=0,1,2} A_K (u, v) \frac{\tau_K}{1 + (\omega\tau_K)^2} + (1 - S^2) P_2(u \cdot v) \frac{\tau_3}{1 + (\omega\tau_3)^2} \tag{1}$$

$u$ and $v$ are the cross-correlated interaction vectors. One has $\tau_3^{-1} = \tau_{int}^{-1} + \tau_{eff}^{-1} = \tau_{int}^{-1} + (4 D_\perp + 2 D_\parallel)$. I regret my oversight in the original report, where I mistook $(4 D_\parallel + 2 D_\perp)$ for $(4 D_\perp + 2 D_\parallel)$.

I have the following reservations with regard to this equation.

(a) The expression for $\tau_3^{-1}$ is valid in the limit where $\tau_{int}^{-1} \gg \tau_{eff}^{-1}$. What is the justification for the validity of this inequality for IDPs?

(b) In the limit of isotropic global motion one has $\sum_{K=0,1,2} A_K (u, v) \frac{\tau_K}{1 + (\omega\tau_K)^2} \rightarrow \frac{\tau_0}{1 + (\omega\tau_0)^2}$. In that limit the authors of mr-2021-35 maintain that $(1 - S^2) P_2(u \cdot v) \rightarrow S_{uv}^2$. Thus, mr-2021-35 predicts the following form for $J_{u,v}(\omega)$ in the limit of isotropic global motion:

$$J_{u,v}(\omega) = \left[ S^2 \frac{\tau_0}{1 + (\omega\tau_0)^2} + S_{uv}^2 \frac{\tau_e}{1 + (\omega\tau_e)^2} \right] = \left[ S^2 \frac{\tau_0}{1 + (\omega\tau_0)^2} + S^2 P_2(u \cdot v) \frac{\tau_e}{1 + (\omega\tau_e)^2} \right] \tag{2}$$

where $\tau_e^{-1} = \tau_{int}^{-1} + 6D_\perp$ and $S_{uv}^2 = S^2 P_2(u \cdot v)$. Yet, in this limit one should have:[2]

$$J_{u,v}(\omega) = P_2(u \cdot v) \left[ S^2 \frac{\tau_0}{1 + (\omega\tau_0)^2} + (1 - S^2) \frac{\tau_e}{1 + (\omega\tau_e)^2} \right] \tag{3}$$

where $\left[ S^2 \frac{\tau_0}{1 + (\omega\tau_0)^2} + (1 - S^2) \frac{\tau_e}{1 + (\omega\tau_e)^2} \right] = J_{u,u}(\omega) = J_{v,v}(\omega)$

(c) The purpose of using the cross-correlation-related expression $\sum_{K=0,1,2} A_K (u, v) \frac{\tau_K}{1 + (\omega\tau_K)^2}$ (eq A.15 of ref 2), instead of the auto-correlation-related expression $\sum_{K=0,1,2} A_K (u) \frac{\tau_K}{1 + (\omega\tau_K)^2}$ (eq A.14 of ref 2) is to allow the effective polar

angle, $\alpha$, in $P_2(u \cdot v) = P_2(\cos \alpha)$, to be relatively large, as it is for the C$^\alpha$–C' spin system (fourth bullet item in mr-2021-35-AC2-supplement). Let us examine this matter. The section after eq A.15 of ref 2 states the following:

in general $\langle P_2(\mu_p(0) \cdot \mu_p(t)) \rangle \neq \langle P_2(\mu_q(0) \cdot \mu_q(t)) \rangle \neq \langle P_2(\mu_p(0) \cdot \mu_q(t)) \rangle P_2(\cos \theta_{pq})$. Thus, the approach used in the text to treat anisotropic overall motion is far from rigorous, although it is expected to be a reasonable approximation when $\theta_{pq}$ is small.

The first two TCFs in this citation yield by Fourier-Laplace transformation the spectral densities $J_{u,u}(\omega)$ and $J_{v,v}(\omega)$ (in the notation of mr-2021-35). "The approach used in the text" refers to the spectral density:

$$J_{u,v}(\omega) = P_2(u \cdot v) \left[ S^2 \sum_{K=0,1,2} A_K(u) \frac{\tau_K}{1 + (\omega\tau_K)^2} + (1 - S^2) \frac{\tau_e}{1 + (\omega\tau_e)^2} \right] \qquad (4)$$

The polar angle, $\alpha$, has to be small enough so that the auto-correlated spectral densities, $J_{u,u}(\omega)$ and $J_{v,v}(\omega)$, are within a good approximation the same. Otherwise the expression in the square brackets of eq 3, which represents the MF spectral density where $J(\omega) = J_{u,u}(\omega) = J_{v,v}(\omega)$, may not be used. Yet, eq 3 is the limit of eq 1 for isotropic global diffusion. For the same reason – inherent equality between $J_{u,u}(\omega)$ and $J_{v,v}(\omega)$ – $\alpha$ has to be small in eq 1. As delineated in my original review report, additional geometric simplifications are inherent in eq 3; they are also inherent in eq 1.

Thus, using eq A14 of ref 2 instead of eq A.15 of ref 2 does not remove the requirement that the polar angle, $\alpha$, be small.

- Eq. (2) (mr-2021-35) is very general. We cannot follow how it might be invalid. In fact, the study of (Tjandra et al. 1996) highlighted as counter-argument features this very expression already in Eq. (2) as well as in the appendix Eq. (A.1).

- The same holds true for the objected expression (3) (mr-2021-35). Eq. (3) is a very general form of a TCF. It only implies that the decay can be modeled as a superposition of exponential decays. All commonly employed analytical models and even MD-extracted TCFs adhere to this general shape. We believe sufficient references have been provided.
- From (2) and (3) follows (4) (mr-2021-35), so we must disagree with the objections raised. With the angle between **u** and **v** fixed, the same expression is also found in (Tjandra et al. 1996) between (A.1) and (A.2).

The sentence in my original report to which the authors refer follows the description of the standard treatment of cross-correlated relaxation for isotropic global diffusion. It reads: "Equations

2–4 do not comply with THIS standard NMR relaxation procedure." The authors interpret "THIS" as "THE". I admit that "do not comply with" is bad phrasing; it should have been "do not represent". By the way, the upper limits in the summations of eqs 3 and 5 of mr-2021-35 should be infinity.

- From (2) to (9) (mr-2021-35) we simply establish the effect of anisotropic diffusion on the TCF which depends on the relative orientations of $\mathbf{u}$ and $\mathbf{v}$. For $H^\alpha H^N$ intraresidual and sequential NOEs, this model has been invoked to rationalize unexpected variations in $\alpha$-Synuclein (Mantsyzov et al. 2014, p. 1281-1282). However, as $H^\alpha H^N$ distances vary with $\varphi$ and $\psi$, the observed effects were ultimately considered to be dominated by distance variations (p. 1286). $C'C^\alpha$CCR would not suffer from this ambiguity, which is why we propose it as an alternative.

  Please see above with regard to Mantsyzov et al.[1]

- We do not understand in what way Eqs. (6) and (7) (mr-2021-35) are confusing. Again, they are taken from the highlighted study of (Tjandra et al. 1996), see Appendix (A.14). The expression suggested by referee #2 makes use of the auto-correlated expression (A.15). We already commented on the possibility to approximate the entire cross-correlated TCF by an auto-correlated TCF on page 6. As the angle between $C'C^\alpha$ and $\sigma_{yy}$ is rather large, we prefer to model $C_{tumb}(t)$ according to (A.14). Referenced in the paper, a different representation has been derived by (Deschamps & Bodenhausen, 2001). Can referee #2 clarify, is the validity of Eqs. (6) and (7) (mr-2021-35) being questioned or the combination with a fourth Lorentzian in Eq. (10)?

  The comment made here refers to the usage of eq A.14 instead of A.15, discussed in item (c) above.

- From (2) to (9) (mr-2021-35) we simply establish the effect of anisotropic diffusion on the TCF which depends on the relative orientations of $\mathbf{u}$ and $\mathbf{v}$. For $H^\alpha H^N$ intraresidual and sequential NOEs, this model has been invoked to rationalize unexpected variations in $\alpha$-Synuclein (Mantsyzov et al. 2014, p. 1281-1282). However, as $H^\alpha H^N$ distances vary with $\varphi$ and $\psi$, the observed effects were ultimately considered to be dominated by distance variations (p. 1286). $C'C^\alpha$CCR would not suffer from this ambiguity, which is why we propose it as an alternative.

  As shown above, eq 1 is not applicable to the spin system $C^\alpha$–C'.

- In addition, to **assess** the effect of isotropic local motions, we simply introduce an additional exponential decay / Lorentzian. As we said in the manuscript "While the fast isotropic motions could be modeled in more detail to better fit the shape of the TCF using e.g. the extended MF approach(Clore et al., 1990) or correlation time distributions(Hsu et al., 2018), we only intend to divide J(0), i.e. the TCF's enclosed area, into contributions with and without orientational biases." (page 6). Eq. (10) (mr-2021-35) is a rough MF-like approximation, $(1-S^2)\tau_3$ is simply the contribution to J(0) attributed to isotropic motions. While in principle arbitrary how this contribution is denoted, $S^2$ and $\tau_{int}$ tend to provide a better "feel" for many. Including additional and/or differently termed isotropic terms would not change the behavior of Q, only their cumulative size is relevant.

The local-motional contribution of $(1 - S^2)\dfrac{\tau_e}{1 + (\omega\tau_e)^2}$ can be justified on the basis of the theory of moments.[3] The local-motional contribution $(1 - S^2)P_2(u \cdot v)\dfrac{\tau_3}{1 + (\omega\tau_3)^2}$ is used allegedly.

Our approximation can be justified from various angles which we have sketched in the paper. While one can argue about the physical meaning of MF-type models, two different approaches connecting $\tau_{int}$ with $\tau_0$ $\tau_1$ $\tau_2$ were highlighted. We agree that the word "coupling" is a poor choice. We were referring to how the factorization (which generally implies **no** dynamic coupling) $C_{tumb}(t)C_{int}(t)$ is handled. It should be noted that already this product form is not strictly applicable in case of anisotropic tumbling. (Kroenke et al. 1998) keep $C_{tumb}(t)$ anisotropic, Eq. (1), which yields a $\tau_3$, $\tau_4$ and $\tau_5$ and retains orientational biases even with $S^2=0$. We follow (Barbato et al. 1992), Eqs. (6a) and (6b) / (Tjandra et al. 1995), sec. Theory, who approximate $\tau_3$ assuming an effective isotropic $C_{tumb}(t)$, see below. $C_{int}(t)$ decays from $P_2(u \cdot v)$ towards $S^2_{uv}$ which for the approximated isotropic tumbling can be expressed as $S^2_{uv} = S^2 P_2(u \cdot v)$ (Ghose et al., 1998), Eq. (19), Appendix, or (Fischer et al. 1997), Eq. (29), which yields the "second term of eq 3".

The reservations expressed in the original report and here concern eq 10 of mr-2021-35. All of the articles cited in the preceding paragraph feature physically well-defined TCFs.

The MF approach of (Halle 2009), which does not assume the factorization of $C_{tumb}(t)C_{int}(t)$, was highlighted as well. The auto-correlated expression for anisotropic diffusion is found in Eq. (2.40). Treatment of cross-correlations are described in section D. Replacing 1 in Eq. (2.40) with $\kappa_{uv} = P_2(\mathbf{u} \cdot \mathbf{v})$, Eq. (2.59), and again approximating $S^2_{uv} = S^2\kappa_{uv} = S^2 P_2(\mathbf{u} \cdot \mathbf{v})$, the expressions of (Tjandra et al. 1996), Eqs. (3), (4) and (6) are obtained. These correspond to Halle's Eq. (2.64) for isotropic internal mobility which relates the cross-correlated TCF to the auto-correlated TCF as suggested by referee #2. The chosen exponential form of $C_{int}(t)$ and the choice of $\tau_{eff}$ can be motivated as above. If the angular dependencies of $C_{tumb}(t)$ are encoded by Eqs. (6) and (7) (mr-2021-35) instead of $S^2_{uv} = S^2 P_2(\mathbf{u} \cdot \mathbf{v})$, Eq. (10) (mr-2021-35) is obtained. The expression suggested by (Ghose et al., 1998), Eq. (7), is very similar but lacks the prefactor $P_2(\mathbf{u} \cdot \mathbf{v})$ for the internal TCF, which is problematic for large angles (e.g. between $C'C^\alpha$ and $\sigma_{yy}$). Regarding the necessity of various assumptions (including frame transformation properties) as well as the general validity of different MF approaches, we find Halle's remarks in VI.A worth highlighting.

Explicitly or implicitly all three MF models (A, B and C) considered by Halle[4] assume statistical independence between the global and internal motions. One may not "choose" arbitrarily an exponential form for $C_{int}(t)$; one has to justify this (e.g., see ref 3). It is shown above why using eqs 6 and 7 does not render eq 1 applicable to arbitrary polar angle, $\alpha$. Equation 7 of Ghose et al.[5] does not lack the factor $P_2(u \cdot v)$; note the summation over $l$ in it, and the evolution of this equation into eq 9 of that article. Halle[4] voices supportive assessment of four specific MF formulae, none claimed to apply to IDPs.

We feel Eq. (10) (mr-2021-35) adheres closely to conventional descriptions of MF-adjusted diffusion anisotropy. Again, we are not claiming this is how an IDP will realistically behave. Rather, what signature would the simplified image of anisotropic tumbling imply? And to what extent could we still detect it if we include faster isotropic motions in a simplified manner?

One cannot attain objectives with inadequate tools.

- "Why should the effective correlation time for global motion be equal to $\tau_2^{-1}$?" It appears referee #2 is mistaken, $\tau_{eff} = (4D_\perp + 2D_\parallel)^{-1} \neq (4D_\parallel + 2D_\perp)^{-1} = \tau_2$. As we referenced

(Barbato et al. 1992, Tjandra et al. 1995), it is calculated from the trace of the diffusion tensor

$$\tau_{eff} = 6D^{-1} = 6\tfrac{1}{3}(D_x + D_y + D_z)^{-1} = 6\tfrac{1}{3}(D_\perp + D_\perp + D_\parallel)^{-1} = (4D_\perp + 2D_\parallel)^{-1}$$

My apologies for this oversight (see above).

**General matters.**

**(2)** A cursory survey of the NMR-based studies of IDPs cited in mr-2021-35 shows that methods examining local features point to a random-coil situation (e.g., see Mantsyzov et al. **2014**). CCR in $^{15}$N–$^{1}$H and $^{13}$C'–$^{13}$C$^{\alpha}$ is a local feature. Obtaining information on long-range "order", in particular "diffusion anisotropy", requires empirical spectral densities comprising statistical elements, to be used in combination with coil libraries and molecular dynamics simulations (e.g., see Mantsyzov et al. **2014**). These elements are absent in mr-2021-35 scheme.

> **(2)** As stated before, the concept of anisotropic tumbling of α-helical and chain-like elements has been invoked in that very same paper (Mantsyzov et al 2014), pages 1281-1282. It was speculated that the local orientation of the spin pairs with respect to the C$^{\alpha}$C$^{\alpha}$ vector (in part) explains the variations of intraresidual and sequential $^{1}$H$^{\alpha1}$H$^{N}$ NOEs.

This is a description of matters addressed in ref 1; it is not a response to comment 2. Please see above for my suggestion in this regard.

**(3)** It is indicated that anisotropic "segmental" motion is targeted. How are these "segments" defined? In other words, how is the second-rank diffusion tensor, *D*, defined?

> **(3)** Following (Mantsyzov et al 2014), we assume that the peptide plane is embedded within the same diffusion tensor. Its unique axis is assumed to lie in the peptide plane such that the sketched edge cases (parallel/perpendicular) are covered. The details are described in the Methods section.

Please see above for related comments/discussion.

**(4)** It is pointed out that NMR relaxation analysis methods applicable to folded proteins are not applicable to IDPs. The model-free (MF) spectral density, a variant of which is suggested here, refers to protein and probe as rigid bodies moving in a statistically independent (decoupled) manner. In IDPs the protein is not rigid and its motion is not decoupled from the motion of the probe. The spectral density suggested here does not reflect these features; rather, it is similar in character to the MF spectral density.

> **(4)** As we mentioned before, we do not expect Eq. (10) (mr-2021-35) to apply in any strict sense. It is indeed the spectral density for MF-like anisotropic tumbling of a (sufficiently) rigid symmetric top. That being said, the general form of Eq. (10), i.e. a weighted sum of Lorentzians, can be expected to describe the TCFs of virtually any protein system in isotropic solution.

I addressed these issued above. Only an infinite sum of Lorentzians describes any system involved in rotational reorientation. Finite sums will be appropriate if they are solutions of physical-relevant models.

**(5)** In the context of item (2) – please note that the "dynamics detectors" method (Smith et al. *Angew. Chem. Int. Ed.* **2017** *56*. 13590), shown to actually surpass MF, comprises statistical elements. Recently it was applied to proteins in solution (Smith et al. *JCP* **2019**, *151*, 034102). The authors might want to check applicability to IDPs.

(5) We agree that fitting experimental relaxation parameters with only few Lorentzians has its limitations. The number of parameters and consequential statistical uncertainties are problems in their own right. In fact, Crawley and Palmer address this issue in this Festschrift (mr-2021-28). In our study, we have referenced different possibilities ranging from correlation time distributions to spectral density visualizations. Again, we are interested in detecting effects of anisotropic dynamics in J(0). We do not intend nor suggest to fit experimental relaxation parameters of IDPs using Eq. (10) (mr-2021-35).

Comment no. 5 refers to the form of the spectral density, not the data-fitting process. One cannot detect actual effects with inappropriate tools.

References
1. Mantsyzov, A. B.; Maltsev, A. S.; Ying, J.; Shen, Y.; Hummer, G.; Bax, A. A Maximum Entropy Approach to the Study of Residue-Specific Backbone Angle Distributions in α-Synuclein, an Intrinsically Disordered Protein. *Protein Science* **2014**, *23*, 1275-1290.

2. Tjandra, N.; Szabo, A.; Bax, A. Protein Backbone Dynamics and $^{15}N$ Chemical Shift Anisotropy from Quantitative Measurement of Relaxation Interference Effects. *J. Am. Chem. Soc*. **1996**, *118*, 6986-6991.

3. Lipari, G.; Szabo, A. Model-Free Approach to the Interpretation of Nuclear Magnetic Resonance Relaxation in Macromolecules. 1. Theory and Range of Validity. *J. Am. Chem. Soc*. **1982**, *104*, 4546-4559.

4. Halle. B. The Physical Basis of Model-Free Analysis of NMR Relaxation Data from Proteins and Complex Fluids. *J. Chem. Phys*. **2009**, *131*, 224507-22.

5. Ghose, R.; Huang, K.; Prestegard, J. H. Measurement of Cross Correlation between Dipolar Coupling and Chemical Shift Anisotropy in the Spin Relaxation of $^{13}C$, $^{15}N$-Labeled Proteins. *J. Magn. Res*. **1998**, *135*, 487-499.

**Summary**: Please provide a cross-correlated spectral density which forgoes the deficiencies pointed out, and relate to the comments made.

---

## Author Comment (AC1)

We agree with the points raised and the arguments are warranted. Our contribution is conceptual in nature and not yet substantiated by experimental results. However, the proposed measurement of longitudinal C'/C$^\alpha$C' CCR is by no means speculative. A possible measurement scheme was sketched and preliminary data for human Ubiquitin is presented in Fig. 3 illustrating that the CCR rates can be measured with sufficient resolution.

Based on the distances alone, CCR to remote carbons (C'C$^\beta$ and C'$_i$C$^\alpha_{i+1}$) would not be completely negligible. However, as the J-couplings 2J(C'C$^\beta$) and 2J(C'$_i$C$^\alpha_{i+1}$) are not resolved, these effects can be expected to average out especially for short and intermediate mixing times. While up to 20% in size of the CCR(C'C$^\alpha$), the +/- CCR(C'C$^\beta$) components contribute to the same line.

Still, we are not at the point to complement our contribution to the Festschrift with definitive experimental data. With the given deadline, the presented concepts and preliminary measurements represent the project's current status. Closely related to Geoffrey Bodenhausen's groundwork, we feel the developed concepts and ideas provide a worthwhile contribution to the Festschrift.

So indeed, by offering a closely connected but differently oriented spin probe complementing the conventional NH$^N$ spin pair, C'/C$^\alpha$C' CCR can provide a "feel" for the local dynamics of the peptide plane and possible differences on a residue-per-residue basis. As the spectral densities are mapped at the same (i.e. zero) frequency, direct comparisons are straightforward.

If the J(0)s and thus Qs vary noticeably and with a persistent pattern over multiple residues and the size of J(0) implies the presence of slower tumbling motions, we would consider this experimental evidence for the previously evoked image of anisotropic tumbling of helical or chain-like elements in IDPs (Mantsyzov et al.). If the J(0)s/Qs deviate from the isotropic case in a systematic and correlated manner, it would hint towards different inherent mobilities for C'/C$^\alpha$C' and N/NH$^N$. This could result from different librational degrees of freedom and/or the effect of flanking $\psi_{i-1}$/$\varphi_i$ flips derived from MD (Salvi et al., Bremi et al.). In the somewhat unexpected scenario of pronounced variation of J(0)s/Qs from residue to residue, it would mean that IDP dynamics are highly anisotropic and heterogeneous. In this case, $^{15}$N relaxation alone could not be expected to capture IDP dynamics in adequate detail. If the time correlation function of such closely connected spin-probes were to decay very differently, it would have substantial structural implications.

If the J(0)s/Qs are very similar and adhere to the isotropic case, it would appear that peptide plane dynamics in IDPs are already well-probed by $^{15}$N relaxation alone and peculiarities observed for sequential H$^\alpha$H$^N$ NOEs should not be attributed to anisotropic dynamics but rather to the additional degree of freedom encoded in $\psi$ (Mantsyzov et al.). In structural terms, diffusive segmental reorientation in IDPs does not correspond to the mental image of isolated helices and chains tumbling in solution.

Of course, the above scenarios might apply differently depending on the protein system.
We can highlight these implications further in the Results & Discussion section.

---

## Author Comment (AC2)

Regarding general matter (**1**), as referee #1 has pointed out as well, our study is indeed predictive and conceptual in nature. In our paper, we want to acknowledge Geoffrey Bodenhausen's contributions to the field by discussing how CCR experiments developed in his group over the years can potentially be used to investigate and characterize local (segmental) dynamics in (partially) disordered proteins.

Let us clarify: We do not claim nor intend to provide a thorough and general description of IDP dynamics. While segmental motions are clearly present in IDPs (Rezaei-Ghaleh et al. 2018, Parigi et al. 2014), it is obvious that the concept of diffusion anisotropy cannot be expected to rigorously apply. Still, this somewhat elusive concept has been invoked in previous studies even for proteins such as α-Synuclein (Mantsyzov et al. 2014). This leads us to the central question of our study: If the segmental motion of IDPs exhibited features/trends associated with anisotropic tumbling, how could we best detect it? Can we define a sensitive and unambiguous experimental measure?

Complementing $NH^N$ CCR, we show that $C'C^\alpha$ CCR allows to probe the peptide plane quite literally from a different angle. The isolated zero frequency components are straightforward to compare and particularly sensitive for larger correlation times, allowing to probe the presence or lack(!) of anisotropic dynamics in IDPs on the local scale of the peptide unit. To assess the combined information content of the CCR rates, we build on the previously invoked and simplified model of a tumbling symmetric top. The "dampening" effect of local motions is approximated as simply as possible using a single exponential decay with equal weight for both $NH^N$ and $C'C^\alpha$. We do not claim this treatment is rigorous nor do we imply that features of this MF-like anisotropic tumbling should be expected for all protein systems. The experiments were conceptualized and designed to assess the presence of features/trends associated with this simplified dynamic model.

In the following paragraphs, we will address the questions and remarks of referee #2. First, we see fit to discuss the formal objections.

- Eq. (2) (mr-2021-35) is very general. We cannot follow how it might be invalid. In fact, the study of (Tjandra et al. 1996) highlighted as counter-argument features this very expression already in Eq. (2) as well as in the appendix Eq. (A.1).

- The same holds true for the objected expression (3) (mr-2021-35). Eq. (3) is a very general form of a TCF. It only implies that the decay can be modeled as a superposition of exponential decays. All commonly employed analytical models and even MD-extracted TCFs adhere to this general shape. We believe sufficient references have been provided.

- From (2) and (3) follows (4) (mr-2021-35), so we must disagree with the objections raised. With the angle between **u** and **v** fixed, the same expression is also found in (Tjandra et al. 1996) between (A.1) and (A.2).

- We do not understand in what way Eqs. (6) and (7) (mr-2021-35) are confusing. Again, they are taken from the highlighted study of (Tjandra et al. 1996), see Appendix (A.14). The expression suggested by referee #2 makes use of the auto-correlated expression (A.15). We already commented on the possibility to approximate the entire cross-correlated TCF by an auto-correlated TCF on page 6. As the angle between $C'C^\alpha$ and $\sigma_{yy}$ is rather large, we prefer to model $C_{tumb}(t)$ according to (A.14). Referenced in the paper, a different representation has been derived by (Deschamps & Bodenhausen, 2001). Can referee #2 clarify, is the validity of Eqs. (6) and (7) (mr-2021-35) being questioned or the combination with a fourth Lorentzian in Eq. (10)?

- From (2) to (9) (mr-2021-35) we simply establish the effect of anisotropic diffusion on the TCF which depends on the relative orientations of **u** and **v**. For $H^\alpha H^N$ intraresidual and sequential NOEs, this model has been invoked to rationalize unexpected variations in α-

Synuclein (Mantsyzov et al. 2014, p. 1281-1282). However, as $H^\alpha H^N$ distances vary with $\varphi$ and $\psi$, the observed effects were ultimately considered to be dominated by distance variations (p. 1286). $C'C^\alpha$ CCR would not suffer from this ambiguity, which is why we propose it as an alternative.

- In addition, to **assess** the effect of isotropic local motions, we simply introduce an additional exponential decay / Lorentzian. As we said in the manuscript "While the fast isotropic motions could be modeled in more detail to better fit the shape of the TCF using e.g. the extended MF approach(Clore et al., 1990) or correlation time distributions(Hsu et al., 2018), we only intend to divide J(0), i.e. the TCF's enclosed area, into contributions with and without orientational biases." (page 6). Eq. (10) (mr-2021-35) is a rough MF-like approximation, $(1-S^2)\tau_3$ is simply the contribution to J(0) attributed to isotropic motions. While in principle arbitrary how this contribution is denoted, $S^2$ and $\tau_{int}$ tend to provide a better "feel" for many. Including additional and/or differently termed isotropic terms would not change the behavior of Q, only their cumulative size is relevant.

Our approximation can be justified from various angles which we have sketched in the paper. While one can argue about the physical meaning of MF-type models, two different approaches connecting $\tau_{int}$ with $\tau_0$ $\tau_1$ $\tau_2$ were highlighted. We agree that the word "coupling" is a poor choice. We were referring to how the factorization (which generally implies **no** dynamic coupling) $C_{tumb}(t)C_{int}(t)$ is handled. It should be noted that already this product form is not strictly applicable in case of anisotropic tumbling. (Kroenke et al. 1998) keep $C_{tumb}(t)$ anisotropic, Eq. (1), which yields a $\tau_3$, $\tau_4$ and $\tau_5$ and retains orientational biases even with $S^2$=0. We follow (Barbato et al. 1992), Eqs. (6a) and (6b) / (Tjandra et al. 1995), sec. Theory, who approximate $\tau_3$ assuming an effective isotropic $C_{tumb}(t)$, see below. $C_{int}(t)$ decays from $P_2(\mathbf{u}\cdot\mathbf{v})$ towards $S^2_{uv}$ which for the approximated isotropic tumbling can be expressed as $S^2_{uv} = S^2 P_2(\mathbf{u}\cdot\mathbf{v})$ (Ghose et al., 1998), Eq. (19), Appendix, or (Fischer et al. 1997), Eq. (29), which yields the "second term of eq 3".

The MF approach of (Halle 2009), which does not assume the factorization of $C_{tumb}(t)C_{int}(t)$, was highlighted as well. The auto-correlated expression for anisotropic diffusion is found in Eq. (2.40). Treatment of cross-correlations are described in section D. Replacing 1 in Eq. (2.40) with $\kappa_{uv} = P_2(\mathbf{u}\cdot\mathbf{v})$, Eq. (2.59), and again approximating $S^2_{uv} = S^2\kappa_{uv} = S^2 P_2(\mathbf{u}\cdot\mathbf{v})$, the expressions of (Tjandra et al. 1996), Eqs. (3), (4) and (6) are obtained. These correspond to Halle's Eq. (2.64) for isotropic internal mobility which relates the cross-correlated TCF to the auto-correlated TCF as suggested by referee #2. The chosen exponential form of $C_{int}(t)$ and the choice of $\tau_{eff}$ can be motivated as above. If the angular dependencies of $C_{tumb}(t)$ are encoded by Eqs. (6) and (7) (mr-2021-35) instead of $S^2_{uv} = S^2 P_2(\mathbf{u}\cdot\mathbf{v})$, Eq. (10) (mr-2021-35) is obtained. The expression suggested by (Ghose et al., 1998), Eq. (7), is very similar but lacks the prefactor $P_2(\mathbf{u}\cdot\mathbf{v})$ for the internal TCF, which is problematic for large angles (e.g. between $C'C^\alpha$ and $\sigma_{yy}$). Regarding the necessity of various assumptions (including frame transformation properties) as well as the general validity of different MF approaches, we find Halle's remarks in VI.A worth highlighting.

We feel Eq. (10) (mr-2021-35) adheres closely to conventional descriptions of MF-adjusted diffusion anisotropy. Again, we are not claiming this is how an IDP will realistically behave. Rather, what signature would the simplified image of anisotropic tumbling imply? And to what extent could we still detect it if we include faster isotropic motions in a simplified manner?

- "Why should the effective correlation time for global motion be equal to $\tau_2^{-1}$?" It appears referee #2 is mistaken, $\tau_{eff} = (4D_\perp + 2D_\parallel)^{-1} \neq (4D_\parallel + 2D_\perp)^{-1} = \tau_2$. As we referenced

(Barbato et al. 1992, Tjandra et al. 1995), it is calculated from the trace of the diffusion tensor

$$\tau_{eff} = 6D^{-1} = 6\,^{1}/_{3}\,(D_x + D_y + D_z)^{-1} = 6\,^{1}/_{3}\,(D_\perp + D_\perp + D_\parallel)^{-1} = (4D_\perp + 2D_\parallel)^{-1}$$

**General matters**

**(2)** As stated before, the concept of anisotropic tumbling of α-helical and chain-like elements has been invoked in that very same paper (Mantsyzov et al 2014), pages 1281-1282. It was speculated that the local orientation of the spin pairs with respect to the $C^\alpha C^\alpha$ vector (in part) explains the variations of intraresidual and sequential $^{1}H^\alpha{}^{1}H^N$ NOEs.

**(3)** Following (Mantsyzov et al 2014), we assume that the peptide plane is embedded within the same diffusion tensor. Its unique axis is assumed to lie in the peptide plane such that the sketched edge cases (parallel/perpendicular) are covered. The details are described in the Methods section.

**(4)** As we mentioned before, we do not expect Eq. (10) (mr-2021-35) to apply in any strict sense. It is indeed the spectral density for MF-like anisotropic tumbling of a (sufficiently) rigid symmetric top. That being said, the general form of Eq. (10), i.e. a weighted sum of Lorentzians, can be expected to describe the TCFs of virtually any protein system in isotropic solution.

**(5)** We agree that fitting experimental relaxation parameters with only few Lorentzians has its limitations. The number of parameters and consequential statistical uncertainties are problems in their own right. In fact, Crawley and Palmer address this issue in this Festschrift (mr-2021-28). In our study, we have referenced different possibilities ranging from correlation time distributions to spectral density visualizations. Again, we are interested in detecting effects of anisotropic dynamics in J(0). We do not intend nor suggest to fit experimental relaxation parameters of IDPs using Eq. (10) (mr-2021-35).

References

Rezaei-Ghaleh, N. *et al. Angewandte Chemie International Edition* **57**, 15262–15266 (2018).

Parigi, G. *et al. J. Am. Chem. Soc.* **136**, 16201–16209 (2014).

Mantsyzov, A. B. *et al. Protein Science* **23**, 1275–1290 (2014).

Tjandra, N., Szabo, A. & Bax, A. *J. Am. Chem. Soc.* **118**, 6986–6991 (1996).

Deschamps, M. & Bodenhausen, G. A. *ChemPhysChem* **2**, 539–543 (2001).

Kroenke, C. D., Loria, J. P., Lee, L. K., Rance, M. & Palmer, A. G. *J. Am. Chem. Soc.* **120**, 7905–7915 (1998).

Barbato, G., Ikura, M., Kay, L. E., Pastor, R. W. & Bax, A. *Biochemistry* **31**, 5269–5278 (1992).

Tjandra, N., Feller, S. E., Pastor, R. W. & Bax, A. *J. Am. Chem. Soc.* **117**, 12562–12566 (1995).

Ghose, R., Huang, K. & Prestegard, J. H. *Journal of Magnetic Resonance* **135**, 487–499 (1998).

Fischer, M. W. F. *et al. J. Am. Chem. Soc.* **119**, 12629–12642 (1997).

Halle, B. *The Journal of Chemical Physics* **131**, 224507 (2009).

---

## Author Comment (AC3)

Authors submitting altogether predictive manuscripts to established journals are typically required to demonstrate applicability. It has been assumed that submissions to the Festschrift issue of MR Discussions honoring Geoffrey Bodenhausen are also subject to this requirement.

As in our response to referee #1, we feel this notion is warranted. Still, with the given deadline, the developed concepts and the preliminary data represent the project's current status. We deem it a worthwhile contribution to the Festschrift.

The first sentence is obvious and the objective of this work is appreciated. The issues at stake include the definition of the "segment" as conceived by Mantsyzov et al.[1] (to which the authors refer), and the physical relevance of the "measure". $C^\alpha$ and C' belong to residue $i - 1$; N and $H^N$ belong to residue $i$. $D_\parallel/D_\perp$ are likely to differ for consecutive $(i - 1) - i$ pairs. How will then the ratios, $D_\parallel/D_\perp$, between the parallel and perpendicular components of the global diffusion tensor, $D$, of the "segment" be determined? The "measure" suggested is ill-defined – see the original review report, and this report. On the most general level – in the Introduction the authors indicate that time-correlation functions (TCFs) developed for folded proteins, which include model-free-type TCFs, are not applicable to intrinsically disordered proteins (IDPs). Yet, they suggest a TCF of this type. With regard to these matters have I suggested considering – or at least discussing – coil libraries and molecular dynamics simulations, e.g., as in Mantsyzov et al.[1]

We feel these two general issues should be treated and highlighted separately. We will refer to these arguments throughout.

Issue 1) Are we justified to invoke the concept of a tumbling symmetric top in the context of IDPs?

As we emphasized before, IDPs cannot be expected to generally adhere to this model. We do not suggest a TCF of this type to model IDPs. We want to probe whether there are anisotropy features/biases in the (segmental) dynamics of (partially) disordered proteins commonly associated with this model. Studies invoking this concept in the context of (partially) disordered proteins were provided ranging from "fully disordered" to partially structured segments. IDPs are not "random coils" but a diverse class of proteins with context specific structural features.

As stated in the manuscript, isolating the J(0) components of $NH^N$ CCR and $C'C^\alpha$ CCR provides a very general measure for the presence of anisotropic dynamics at the scale of the peptide plane even without a specific dynamic model in mind. Since we can identify the isotropic case, we can also detect deviations thereof. We only consider the simplistic and qualitative definition of a tumbling "segment" previously asserted by (Mantsyzov et al. 2014) to assess the sensitivity of Q. As emphasized in the Introduction, we are agnostic about the relevance/applicability of this simplified representation. Q could likely rule out this concept in many/most cases, it depends on the specific signature. The experiments were designed to probe the presence of anisotropic "slow" motions associated with segmental tumbling. If the J(0)s of the two spin pairs are similar, peptide plane dynamics in IDPs would appear to be isotropic and well-probed by conventional $^{15}$N relaxation. If the obtained J(0)s exhibit pronounced differences, the source of anisotropy should certainly be investigated in further detail (preferably in conjunction with MD simulations). As IDPs are a diverse family of proteins, the results will necessarily depend on the system under investigation.

We do not see any inherent contradiction in assessing and illustrating the sensitivity of Q using the previously invoked concept of a symmetric top. The limited validity of this model for IDPs is emphasized throughout the paper. We can certainly emphasize these considerations and further clarify the exemplary nature of this model in the manuscript.

Issue 2) Does our proposed spectral density represent CCR of a tumbling symmetric top? Is the approximate MF-like inclusion of local motions properly handled?

Based on the objections thus far, we deem our description of the tumbling symmetric top appropriate as it is. The apparent deficiencies are addressed in detail below.

In the following, we will address the open questions and comments:

As stated in our previous response and in the original manuscript, we do not suggest that any parameters such as $D_\parallel/D_\perp$ could be "determined" as the dynamic model is too simplified. We are only comparing the J(0)s, we do not claim that solving the inverse problem, i.e. extraction of the (hypothetical) diffusion tensor, was possible or intended. That being said, why would $D_\parallel/D_\perp$ be likely to differ for consecutive peptide planes if they tumble in a concerted fashion? We would not rule out this effect, but we would speculate that the relative orientations of the peptide planes are a more relevant source of uncertainty.

MD simulations are highlighted throughout the manuscript, but we can further emphasize their potential. In the authors' opinions, MD simulations are the ideal "dynamic model" for IDPs. If the CCR rates turn out to be "heterogeneous enough", we will certainly include simulations in a future study. Limited to (φ,ψ)-space, like in (Mantsyzov et al. 2014), coil libraries can be suitable priors for structural inference (Kauffmann et al. 2021). To investigate dynamics of consecutive residues, we tend to prefer MD simulations over the static and generic nature of coil priors.

(a) The expression for $\tau_3^{-1}$ is valid in the limit where $\tau_{int}^{-1} \gg \tau_{eff}^{-1}$. What is the justification for the validity of this inequality for IDPs?

See Issue 1, we are not modeling the dynamics of IDPs, we are considering a MF-like symmetric top.

(b) In the limit of isotropic global motion one has $\sum_{K=0,1,2} A_K(u,v) \frac{\tau_K}{1+(\omega\tau_K)^2} \to \frac{\tau_0}{1+(\omega\tau_0)^2}$.

Where does the angular relation between $u$ and $v$ go? As pointed out by referee #2 in the context of (Ghose et al. 1998) as well, in the limit of isotropic global motion one has $\Sigma_{K=0,1,2} A_K(u,v) \frac{\tau_K}{1+(\omega\tau_K)^2} \to P_2(u \cdot v) \frac{\tau_0}{1+(\omega\tau_0)^2}$. As the index "0" might be confusing for some readers, we note that as $D_\parallel/D_\perp = 1$, $\tau_0 = \tau_1 = \tau_2 = \tau_{tumb} = (6D)^{-1}$.

In that limit the authors of mr-2021-35 maintain that $(1-S^2) P_2(u \cdot v) \to S_{uv}^2$.

This is not correct. In that limit we state that $S_{uv}^2 \to P_2(u \cdot v)S^2$ such that
$P_2(u \cdot v) - S_{uv}^2 \to (1-S^2)P_2(u \cdot v)$

$$J_{u,v}(\omega) = \left[ S^2 \frac{\tau_0}{1+(\omega\tau_0)^2} + S_{uv}^2 \frac{\tau_e}{1+(\omega\tau_e)^2} \right] = \left[ S^2 \frac{\tau_0}{1+(\omega\tau_0)^2} + S^2 P_2(u \cdot v) \frac{\tau_e}{1+(\omega\tau_e)^2} \right] \qquad (2)$$

This is not the isotropic limit of Eq 1 (Eq 10 mr-2021-35). As stated above, $A_K(u,v)$ sums up to $P_2(u \cdot v)$ and we are not sure why the "auto" $S^2$ and the "cross" $S_{uv}^2$ became mixed liked this. The isotropic limit of Eq 1 (Eq 10 mr-2021-35) is exactly as required:

$$J_{u,v}(\omega) = P_2(u \cdot v) \left[ S^2 \frac{\tau_0}{1+(\omega\tau_0)^2} + (1-S^2) \frac{\tau_e}{1+(\omega\tau_e)^2} \right] \qquad (3)$$

If $S_{uv}^2$ was used throughout, we would instead obtain

$$J_{u,v}(\omega) = S_{uv}^2 \frac{\tau_0}{1 + (\omega\tau_0)^2} + (P_2(u \cdot v) - S_{uv}^2) \frac{\tau_e}{1 + (\omega\tau_e)^2}$$

(Ghose et al. 1998) Eq 9, as later referred to by referee #2, with the angular relations now contained in $S_{uv}^2$.

This isotropic limit is both contained in Eq 11 (mr-2021-35) and illustrated in Fig. 4 (mr-2021-35). As mentioned in the manuscript, it can be seen that the limit of fully isotropic local motions and isotropic tumbling coincides at around 0.3.

Thus, using eq A14 of ref 2 instead of eq A.15 of ref 2 does not remove the requirement that the polar angle, α, be small.

With the validity of A.14 (Tjandra et al. 1996) now established, we can agree that it represents the analytic solution for a tumbling rigid symmetric top. This is an improvement over the approximation via the auto-correlated TCF. Thus, for $S^2 = 1$, Eq. 1 (Eq. 10 mr-2021-35) is exact. For $S^2 = 0$, it is also exact. As established above, for isotropic tumbling, $D_\parallel/D_\perp = 1$, it also follows MF-type expressions as above and referenced in the manuscript/response.

In essence, Eq. 1 (Eq. 10 mr-2021-35) weights/mixes the two exact solutions via $S^2$. This approximation becomes better the closer $S^2$ is to 0 or 1, the closer $D_\parallel/D_\perp$ is to 1 and/or the closer $u \cdot v$ is to 1. We cannot follow at what size and combination of $u \cdot v$, $S^2$ and $D_\parallel/D_\perp$ the approximation becomes entirely unreasonable especially when considering that the only "large" angle is between C'C$^\alpha$ and σ$_{yy}$ which contributes with smaller weight in terms of both size (σ$_{yy}$-σ$_{zz}$) and the initial value of $P_2(u \cdot v)$ (as highlighted in the manuscript). $J_{C'C\alpha,xx}(0)$ is clearly the dominant component.

We cannot see how the value of $u \cdot v$ for C'C$^\alpha$ and σ$_{yy}$ might break the (anyways approximate) interpolation between rigid anisotropic tumbling and fully isotropic motions presented in Fig. 4 (mr-2021-35).

The sentence in my original report to which the authors refer follows the description of the standard treatment of cross-correlated relaxation for isotropic global diffusion. It reads: "Equations 2–4 do not comply with THIS standard NMR relaxation procedure." The authors interpret "THIS" as "THE". I admit that "do not comply with" is bad phrasing; it should have been "do not represent". By the way, the upper limits in the summations of eqs 3 and 5 of mr-2021-35 should be infinity.

This is not an issue of semantics. We still do not see how Eqs. 2-4 (mr-2021-35) do not represent very general properties of TCFs encountered in solution-state NMR. How are MF-like TCFs not contained within these boundaries?

Clearly Eq. 3 is less general than Eq. 2, one could generalize further and consider an infinite sum. We never introduced Eq. 3 as the most general description, we only responded to the claim that it does not comply with MF-type TCFs.

For most practical purposes, we do not see how the difference between large N and infinity might result in substantial discrepancies both in the context of our manuscript and with respect to TCF shapes of proteins in isotropic solution. That being said, we do value the benefits of continuous

notation. Before introducing Eq. 3, we specifically reference the concept of correlation time distributions (mr-2021-35 page 4).

been derived by (Deschamps & Bodenhausen, 2001). Can referee #2 clarify, is the validity of Eqs. (6) and (7) (mr-2021-35) being questioned or the combination with a fourth Lorentzian in Eq. (10)?

The comment made here refers to the usage of eq A.14 instead of A.15, discussed in item (c) above.

In the first response of referee #2, we got the impression that the general notation and validity of A.14 (Tjandra et al. 1996) was being questioned. We apologize if we misunderstood.

- From (2) to (9) (mr-2021-35) we simply establish the effect of anisotropic diffusion on the TCF which depends on the relative orientations of **u** and **v**. For $H^{\alpha}H^{N}$ intraresidual and

Please see above with regard to Mantsyzov et al.[1]

We were responding to the following statement, referring to Eqs 2-9 (mr-2021-35) as problematic:

Based on eqs 2–9, shown above to be problematic, the authors obtain the spectral density:

$$J_{u,v}(\omega) = S^2 \sum_{K=0,1,2} A_K(u,v) \frac{\tau_K}{1+(\omega\tau_K)^2} + (1-S^2) \sum_{K=0,1,2} A_K(u,v) \frac{\tau_3}{1+(\omega\tau_3)^2} \quad (3)$$

In case this led to confusion, it should be highlighted that this is not the proposed spectral density (Eq 10 mr-2021-35). We considered this as a transcription error.

More generally, see Issues 1 and 2.

As shown above, eq 1 is not applicable to the spin system $C^{\alpha}$–C'.

As discussed above, we do not agree with this statement. In addition, the highlighted TCF of (Ghose et al. 1998) includes a similar spin system, namely C'–N. Arguably, the angular relations of C' CSA – C'N DP CCR are not too different from C' CSA – C'C$^{\alpha}$ DP CCR. More details below.

The local-motional contribution of $(1-S^2)\frac{\tau_e}{1+(\omega\tau_e)^2}$ can be justified on the basis of the theory of moments.[3] The local-motional contribution $(1-S^2)P_2(u \cdot v)\frac{\tau_3}{1+(\omega\tau_3)^2}$ is used allegedly.

We discussed in multiple paragraphs how the isotropic contribution can be justified. It is part of the reason we denote it as such as it provides a better "feel" for many. Still, we find the contribution of $P_2(u \cdot v)(1-S^2)\frac{\tau_3}{1+(\omega\tau_3)^2}$ on J(0) worth highlighting in purely mathematical terms.

The reservations expressed in the original report and here concern eq 10 of mr-2021-35. All of the articles cited in the preceding paragraph feature physically well-defined TCFs.

We were referring and contextualizing with respect to the question of referee #2: "What physical framework could possibly justify the second term of eq 3?", i.e. $P_2(u \cdot v)(1-S^2)\frac{\tau_3}{1+(\omega\tau_3)^2}$

The articles were cited to explain all components, i.e. the factor $P_2(u \cdot v)$, the use of an auto-correlated order parameter $S^2$ and the concept of the effective isotropic tumbling time.

Explicitly or implicitly all three MF models (A, B and C) considered by Halle[4] assume statistical independence between the global and internal motions. One may not "choose" arbitrarily an exponential form for $C_{int}(t)$; one has to justify this (e.g., see ref 3). It is shown above why using eqs 6 and 7 does not render eq 1 applicable to arbitrary polar angle, α. Equation 7 of Ghose et al.[5] does not lack the factor $P_2(u \cdot v)$; note the summation over $l$ in it, and the evolution of this equation into eq 9 of that article. Halle[4] voices supportive assessment of four specific MF formulae, none claimed to apply to IDPs.

Generally, see Issues 1 and 2.
Specifically:

As Halle's MF-B (Halle 2009) does not prespecify the form of $C_{int}(t)$, the exponential form requires commentary. We specifically referred to the previous references to justify it.

As stated above, the effect of summation over l is not the issue. We explicitly referred to the **internal TCF** of (Ghose et al. 1998) Eq. 9 (using the previous notation):

$$J_{u,v}(\omega) = S^2 \sum_{K=0}^{2} A_k(u,v) \frac{\tau_k}{1 + (\omega\tau_k)^2} + (1 - S^2)\frac{\tau_e}{1 + (\omega\tau_e)^2} \qquad (a)$$

Note that the index "x" for "cross" in the order parameters has been omitted for clarity. (Ghose et al. 1998) justify the above expression as follows "In order to obtain expressions analogous to conventional Lipari–Szabo theory, we express the auto- and cross-correlation spectral density functions using effective order parameters." (page 488).

Defined to be analogous to the conventional "auto" order parameters, the "effective order parameters" lie between 0 and 1. This can be seen in two ways: In the fully rigid tumbling limit $J_{u,v}(\omega) = \sum_{k=0}^{2} A_k(u,v)\frac{\tau_k}{1+(\omega\tau_k)^2}$, thus $S^2 = 1$. It can also be seen from the initial value 1 of the internal TCF.

We cannot follow the "evolution of this equation into eq 9 of that article". For isotropic tumbling, $\Sigma_{K=0,1,2}A_K(u,v)\frac{\tau_K}{1+(\omega\tau_K)^2} \rightarrow P_2(u \cdot v)\frac{\tau_0}{1+(\omega\tau_0)^2}$, we would obtain:

$$J_{u,v}(\omega) = S^2 P_2(u \cdot v)\frac{\tau_0}{1 + (\omega\tau_0)^2} + (1 - S^2)\frac{\tau_e}{1 + (\omega\tau_e)^2} \qquad (b)$$

This is not the isotropic limit of Eq. 9 of (Ghose et al.) which reads

$$J_{u,v}(\omega) = S_{uv}^2 \frac{\tau_0}{1 + (\omega\tau_0)^2} + (P_2(u \cdot v) - S_{uv}^2)\frac{\tau_e}{1 + (\omega\tau_e)^2} \qquad (c)$$

with the angular relations encoded by $S_{uv}^2$. Using $S_{uv}^2 = P_2(u \cdot v)S^2$, we obtain the familiar

$$J_{u,v}(\omega) = P_2(u \cdot v)(S^2 \frac{\tau_0}{1 + (\omega\tau_0)^2} + (1 - S^2)\frac{\tau_e}{1 + (\omega\tau_e)^2}) \qquad (d)$$

Importantly, we do not think (Ghose et al. 1998) are implying Eq 7 (a) evolves into Eq 9 (c). Eq 7 (a) is defined purely by analogy. Arguably, the index "x" for "cross" is used in a somewhat confusing manner. If the order parameters in Eq 7 (a) were interpreted as "proper" cross-type, one would

encounter different issues. In the fully rigid tumbling limit, the angles would be accounted for twice, i.e. it would decay from $P_2(u \cdot v)^2$ while the internal TCF would still decay from 1. One would not end up with Eq. 9 (c).

It can be seen how this not too problematic for smaller angles, but as we stated in our first response, we feel that the lack of $P_2(u \cdot v)$ for the **internal** TCF is problematic for larger angles. By having the internal TCF decay from $P_2(u \cdot v)$ towards $S_{uv}^2 = P_2(u \cdot v)S^2$, Eq. 10 (mr-2021-35) is obtained.

To be clear, we feel the expressions of (Ghose et al. 1998) are still reasonable. Analogies are a good way to make the features of an (anyways approximate) dynamic model more relatable.

One cannot attain objectives with inadequate tools.

Regarding the sensitivity assessment and the dynamic model, we feel it is adequate, see Issues 1 and 2. The isolated J(0) components of NH$^N$ CCR and C'C$^\alpha$ CCR probe the motions of the shared peptide plane at the same frequency. Thus, they are straightforward to compare and the expectation for isotropic motions can easily be specified. To us this is an adequate tool for detecting the presence of anisotropic dynamics.

Comment no. 5 refers to the form of the spectral density, not the data-fitting process. One cannot detect actual effects with inappropriate tools.

We apologize for the misunderstanding. We already commented on the concept of correlation time distributions, which the approach appears to build on. However, it puts an interesting twist on it by emphasizing the timescales the relaxation parameters are most sensitive to. For now, we feel our proposed (spectral density mapping like) protocol should be appropriate for detecting actual effects without specifying the form of the spectral density. Regarding the spectral density used to assess our protocol, see Issues 1 and 2. The references are appreciated and might prove useful in future applications when it comes to interpreting the observed effects in structural/dynamical terms.

Comments (2), (3) and (4) have been addressed above.

References

Mantsyzov, A. B. *et al.* A maximum entropy approach to the study of residue-specific backbone angle distributions in α-synuclein, an intrinsically disordered protein. *Protein Science* **23**, 1275–1290 (2014).

Kauffmann, C., Zawadzka-Kazimierczuk, A., Kontaxis, G. & Konrat, R. Using Cross-Correlated Spin Relaxation to Characterize Backbone Dihedral Angle Distributions of Flexible Protein Segments. *ChemPhysChem* **22**, 18–28 (2021).

Ghose, R., Huang, K. & Prestegard, J. H. Measurement of Cross Correlation between Dipolar Coupling and Chemical Shift Anisotropy in the Spin Relaxation of13C,15N-Labeled Proteins. *Journal of Magnetic Resonance* **135**, 487–499 (1998).

Tjandra, N., Szabo, A. & Bax, A. Protein Backbone Dynamics and 15N Chemical Shift Anisotropy from Quantitative Measurement of Relaxation Interference Effects. *J. Am. Chem. Soc.* **118**, 6986–6991 (1996).

Halle, B. The physical basis of model-free analysis of NMR relaxation data from proteins and complex fluids. *The Journal of Chemical Physics* **131**, 224507 (2009).

---

## Author Response (AR1)

**Final response**

First and foremost, we want to thank referee #1 and #2 for their valuable insights. The – at times lively – discussion was appreciated. In this final response, we will shortly summarize some of the arguments laid out in the previous responses (mr-2021-35-AC1, mr-2021-35-AC2, mr-2021-35-AC3, mr-2021-35-AC4). We will mostly refer to the changes made to the manuscript, we will not recapitulate the previous exchanges in detail.

Referee #1

(1) *"For example, in the case of an IDP where little residual structure is present and conformations are in rapid flux, and where diffusion anisotropy could well be present on a per-residue basis, will this method be able to detect this qualitatively? Will a per-residue "feel" (even qualitative) of the dynamics be attainable? Could the authors present some experimental data to try to answer these questions, or the general question of what insights we might be able to obtain practically."*

As discussed in our initial response, this is indeed a fair point. While we cannot provide detailed experimental results at this stage, we extended the Results & Discussion section by a qualitative discussion of possible scenarios to exemplify the attainable information content (page 11-12).

(2) *"Can the authors comment on whether interactions involving other 13C spins in a U-13C labeled IDP would be predicted to influence the extracted cross-correlation rates, especially in cases where rates are small."*

We included the considerations of our inital response in the Methods section. The expected CCR effects of remote carbons are discussed on page 9.

Referee #2

*"(1) This study is totally predictive in nature. No validation, and/or examples illustrating actual applicability are provided."*

As previously stated in our two responses and in the response (1) to referee #1, this notion is warranted. Our study is indeed predictive and conceptual in nature. With the given deadline, the presented concepts and preliminary measurements represent the project's current status. We deem it a worthwhile conribution to the Festschrift.

*"(2) A cursory survey of the NMR-based studies of IDPs cited in mr-2021-35 shows that methods examining local features point to a random-coil situation (e.g., see Mantsyzov et al. 2014). CCR in $^{15}N$-$^{1}H$ and $^{13}C'$-$^{13}C^{\alpha}$ is a local feature. Obtaining information on long-range "order", in particular "diffusion anisotropy", requires empirical spectral densities comprising statistical elements, to be used in combination with coil libraries and molecular dynamics simulations (e.g., see Mantsyzov et al. 2014). These elements are absent in mr-2021-35 scheme."*

As previously discussed, the study of (Mantsyzov et al. 2014) invokes the concept of anisotropic diffusion of extended chain segments to rationalize unexpected variations in intraresidual and sequential $H^{\alpha}H^{N}$ NOEs, which are indeed local features. Clearly, long range "order" effects can

manifest in local spin probes. We agree that detailed mechanistic insights can only be obtained by including additional tools such as MD simulations. We are not implying that the spectral density – used to assess the information content of the CCR rates – can or should be used to "determine" a hypothetical diffusion tensor.

We adjusted the phrasing thorughout the manuscript to clarify the general nature of the measure Q as means to detect the presence of anisotropic dynamics, emphasizing the sensivity to slow segmental motions. We further clarify the toy nature of the tumbling symmetric top model and highlight the importance of MD simulations to rationalize experimental CCR rates in IDPs (e.g. page 11-12). As the concept of diffusion anisotropy might be too specific/evocative at times, the phrasing has been adjusted throughout.

*"(3) It is indicated that anisotropic "segmental" motion is targeted. How are these "segments" defined? In other word, how is the second-rank diffusion tensor, D, defined?"*

As described in the previous responses, we adopt the simplistic image of (Mantsyzov et al. 2014), assuming that the main axis of the axially symmetric diffusion tensor lies in the peptide plane. The orientation of the diffusion tensor with respect to the peptide plane is varied as described in the Methods section and covers the cases of idealized α-helices and extended chains as described by (Mantsyzov et al. 2014).

We now highlight the model of (Mantsyzov et al. 2014) in the introduction of the Theory section, page 4.

*"(4) It is pointed out that NMR relaxation analysis methods applicable to folded proteins are not applicable to IDPs. The model-free (MF) spectral density, a variant of which is suggested here, refers to protein and probe as rigid bodies moving in a statistically independent (decoupled) manner. In IDPs the protein is not rigid and its motion is not decoupled from the motion of the probe. The spectral density suggested here does not reflect these features; rather, it is similar in character to the MF spectral density."*

As discussed in more detail in the previous responses, spectral densities of IDPs indeed cannot be expected to adhere to the simplified model underlying the MF-like spectral density. The spectral density is not intended to describe the dynamics of IDPs in any general sense, it is only used to assess the features previously asserted by (Mantsyzov et al. 2014). The experimental measure Q is sensitive to the presence of anisotropic dynamics independently of its source.

We further highlight the toy nature of the MF-like spectral density throughout the manuscript. The limited validity of the model and the importance of MD simulations for mechanistic interpretations are now further emphasized in the Results & Discussion section (page 11-12).

*"(5) In the context of item (2) – please note that the "dynamics detectors" method (Smith et al. Angew. Chem. Int. Ed. 2017 56. 13590), shown to actually surpass MF, comprises statistical elements. Recently it was applied to proteins in solution (Smith et al. JCP 2019, 151, 034102). The authors might want to check applicability to IDPs."*

As previously discussed, this approach is indeed an interesting way to characterize/analyze the spectral densities in further detail. The references are now included and contextualized in the Theory and Results & Discussion sections.

*Mathematical and physical objections regarding the MF-like spectral density*

While the initial mathematical objections could apparently be clarified during the previouses exchanges, the physical meaning of the order paramater and the use of an (apparent) auto-correlated TCF remained an open issue.

We reworked the Theory section to contextualize the proposed spectral density as a simplifciation of the formalism proposed by (Ghose et al. 1998) and generalized by (Vögeli and Yao 2009). Both the auto-correlated form of the TCF and the meaning of the generalized order paramaters are now described in the newly added Appendix. The proposed spectral density is highlighted as a simplification. In the semi-qualitative context of Fig. 4., we consider the use of a single generalized order paramater as a sensible approximation. The more thorough way of handling internal motions is described in the Appendix as well. Parallels to the MF formalisms of (Lipari and Szabo 1982) and (Halle and Wennerström 1981) are illustrated.

References

Mantsyzov, A. B. *et al.* A maximum entropy approach to the study of residue-specific backbone angle distributions in α-synuclein, an intrinsically disordered protein. *Protein Science* **23**, 1275–1290 (2014).

Ghose, R., Huang, K. & Prestegard, J. H. Measurement of Cross Correlation between Dipolar Coupling and Chemical Shift Anisotropy in the Spin Relaxation of13C,15N-Labeled Proteins. *Journal of Magnetic Resonance* **135**, 487–499 (1998).

Vögeli, B. & Yao, L. Correlated dynamics between protein HN and HC bonds observed by NMR cross relaxation. *Journal of the American Chemical Society* **131**, 3668–3678 (2009).

Lipari, G. & Szabo, A. Model-Free Approach to the Interpretation of Nuclear Magnetic Resonance Relaxation in Macromolecules. 1. Theory and Range of Validity. *Journal of the American Chemical Society* **104**, 4546–4559 (1982).

Halle, B. & Wennerström, H. Interpretation of magnetic resonance data from water nuclei in heterogeneous systems. *J. Chem. Phys.* **75**, 1928–1943 (1981).